


# Lower-cost eddy covariance for $CO_2$ and $H_2O$ fluxes over grassland and agroforestry

Justus G. V. van Ramshorst[1,2], Alexander Knohl[1,3], José Ángel Callejas-Rodelas[1], Robert Clement[2,4], Timothy C. Hill[4], Lukas Siebicke[1], and Christian Markwitz[1]

[1]Bioclimatology, Faculty of Forest Sciences and Forest Ecology, University of Göttingen, Büsgenweg 2, 37077 Göttingen, Germany
[2]Quanterra Systems Ltd., Centenary House, Peninsula Park, Exeter EX2 7XE, UK
[3]Centre of Biodiversity and Sustainable Land Use (CBL), University of Göttingen, Göttingen, Germany
[4]Department of Geography, College of Life and Environmental Sciences, University of Exeter, UK

**Correspondence:** Justus van Ramshorst (justus.vanramshorst@uni-goettingen.de)

**Abstract.** Eddy covariance (EC) measurements can provide direct and non-invasive ecosystem measurements of the exchange of energy, water ($H_2O$) and carbon dioxide ($CO_2$). However, conventional eddy covariance (CON-EC) setups (ultrasonic anemometer and infrared gas analyser) can be expensive, which recently led to the development of lower-cost eddy covariance (LC-EC) setups. In the current study we test the performance of a LC-EC setup for $CO_2$ and $H_2O$ flux measurements at

an agroforestry and adjacent grassland site in a temperate ecosystem in northern Germany. The closed-path LC-EC setup was compared with a CON-EC setup using an enclosed-path gas analyser (LI-7200, LI-COR Inc., Lincoln, NE, USA). The LC-EC $CO_2$ fluxes were lower compared to CON-EC by 7-13% ($R^2$ = 0.91-0.95) and the latent heat fluxes were higher by 2-3% in 2020 and 23% in 2021 ($R^2$ = 0.84-0.90). The large difference between latent heat fluxes in 2021, seems to be a consequence of the lower LE fluxes measured by the CON-EC. Due to the slower response sensors of the LC-EC setup, the (co)spectra

of the LC-EC were more attenuated in the high-frequency range compared to the CON-EC. This stronger attenuation of the LC-EC requires a larger spectral correction and as a consequence larger differences between spectral correction factors of different spectral correction methods. At the agroforestry site where the flux tower was taller compared to the grassland, the attenuation was lower, because the cospectrum peak and energy-containing eddies shift to lower frequencies which the LC-EC can measure. With the LC-EC and CON-EC systems was shown that the agroforestry site had a 2.3 times higher carbon uptake

compared to the grassland site and both had an equal evapotranspiration when simultaneously measured for one month. Our results show that LC-EC has the potential to measure EC fluxes at various land-use systems for approximately 25% of the costs of a CON-EC system.

## 1   Introduction

The world is experiencing global warming and climate change, due to the increased greenhouse gas (GHG) concentrations in

the atmosphere (IPCC, 2021). Reducing carbon-dioxide ($CO_2$) and other GHG emissions can minimize these effects (Griscom et al., 2017; Anderson et al., 2019; IPCC, 2021). Mitigating $CO_2$ emissions with Nature-based Climate (management) Solutions



(NbCS) is seen as a fairly rapid and low-cost solution, which meanwhile can provide environmental co-benefits (Griscom et al., 2017; Anderson et al., 2019). By applying NbCS, potentially more carbon can be captured and stored compared to the previous land use or conventional management (Zomer et al., 2016; Griscom et al., 2017; Anderson et al., 2019).

Agroforestry (AF) is a solution to mitigate carbon emissions and at the same time provide resilient agriculture adapted for climate change (Schoeneberger et al., 2012; Smith et al., 2013; Cardinael et al., 2021). In agroforestry systems, trees or tree strips are interleaved with either annual rotating crops or perennial grassland. Agroforestry systems can create a more favorable local microclimate and soften the effect of hot and dry summers (Schoeneberger et al., 2012; Smith et al., 2013; Cardinael et al., 2021). Furthermore, agroforestry can improve the biodiversity (Jose, 2009; Torralba et al., 2016) and reduce
soil erosion (Schoeneberger et al., 2012; van Ramshorst et al., 2022). Nevertheless, robustly validating estimations and models of the carbon sequestration potential by agroforestry and other NbCS is not straightforward and is time and labor intensive (Griscom et al., 2017; Novick et al., 2022). Eddy covariance (EC) can provide solid and independent measurements to validate the carbon uptake of the entire ecosystem (Hemes et al., 2021; Novick et al., 2022; Wiesner et al., 2022).

Eddy Covariance is a non-invasive technique to directly measure the net land-atmosphere exchange (flux) of energy, water
($H_2O$), $CO_2$ and other GHGs over an area of up to several hectares (Baldocchi, 2003; Lee et al., 2005; Baldocchi, 2008). Currently, several global networks of EC towers provide essential data quantifying the net carbon exchange (Sabbatini et al., 2018; Pastorello et al., 2020) and associated climate and land use change impacts, for of a variety of ecosystems. However, conventional EC (CON-EC) systems are expensive and therefore the number of observations are often limited to primary ecosystems and users who can afford EC (Schimel et al., 2015; Hill et al., 2017; Baldocchi, 2020). Consequently, a small number of EC
towers are generally used to represent an ecosystem, which could raise concerns regarding the spatial representativeness of flux measurements, especially when the ecosystem is heterogeneous (Hill et al., 2017; Cunliffe et al., 2022).

Recently, several lower-cost eddy covariance (LC-EC) gas analysers have been developed to provide cheaper but still accurate and robust measurements for $H_2O$ fluxes (Markwitz and Siebicke, 2019), and the combination of $CO_2$ and $H_2O$ fluxes (Hill et al., 2017; Cunliffe et al., 2022). These LC-EC systems use more economical parts and have slower-response sensors,
which leads to a price reduction compared to CON-EC. The LC-EC system of the current study has a price reduction of approximately 75% compared to CON-EC (Cunliffe et al., 2022). Using slower-response sensors, however, leads to an increased loss of high-frequency signal and accordingly this leads to an increased measurement uncertainty (Hill et al., 2017; Markwitz and Siebicke, 2019; Cunliffe et al., 2022). Nevertheless, previous field comparison of LC-EC systems provided flux measurements in agreement with a CON-EC setup (Hill et al., 2017; Markwitz and Siebicke, 2019; Cunliffe et al., 2022).

The additional loss of the high-frequency signal of LC-EC setups increases the importance of the spectral correction applied (Mauder and Foken, 2006; Reitz et al., 2022). Spectral corrections are part of the EC methodology, and these are applied to compensate for the spectral attenuations which are inevitable (Massman and Clement, 2005; Emad, 2023). The magnitude of spectral losses are for example depended on the response time of sensors and the EC system as a whole (Leuning and Moncrieff, 1990; Massman and Lee, 2002; Polonik et al., 2019), the measurement height of the EC tower (Moncrieff et al., 1997; Reitz
et al., 2022), the length and diameter of the tubing when present (Leuning and Moncrieff, 1990; Massman, 1991), the flow rate and flow regime inside the tube (Leuning and Moncrieff, 1990; Massman, 1991), and the absorption and desorption of water





molecules inside the tubing (Massman, 1991; Ibrom et al., 2007; Polonik et al., 2019). Furthermore, there are many different spectral correction methods available, each with their own assumptions, which is an additional source of uncertainty in itself (Polonik et al., 2019; Reitz et al., 2022; Emad, 2023).

In the current study we tested LC-EC setups over a temperate grassland and an adjacent alley cropping agroforestry grassland near Hanover in Germany. The objectives of this paper are to (i) perform a technical characterisation of the LC-EC setup relative to CON-EC in a temperate ecosystem setting, (ii) investigate the effect of the spectral correction method applied, and (iii) present the first application of LC-EC over a grassland and alley cropping agroforesty grassland.

## 2    Methods

### 2.1    Site characterisation

The current study took place at a grassland site in Mariensee, Lower Saxony, Germany (52° 33′ 52.3″ N, 9° 27′ 51.2″ E) (Figure 1). The 7 ha grassland site includes three parallel north-south orientated willow tree strips of approximately 6.5 m height during the time of study (Markwitz et al., 2020). Mowing of the non-grazed grassland was done twice a year, once in summer and once in autumn. The soil consists of Histosol and Anthrosol and has a bulk density of 1.28 kg m$^{-3}$ (Beule et al., 2019; Markwitz et al., 2020).

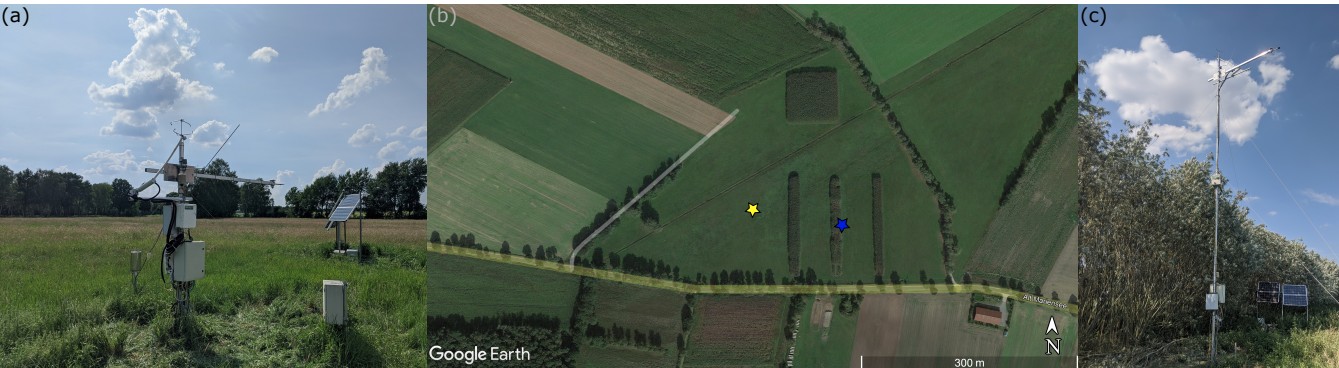

**Figure 1.** a) The Mariensee grassland tower west of the tree strips in June 2020. Photo facing west (Photo by Justus van Ramshorst). b) Satellite image from the Mariensee site, with the yellow and blue star indicating the location of the grassland tower and of the agroforestry tower, respectively (Google Earth, © Google 2022). c) The Mariensee agroforestry tower east of the central tree strip in August 2020. Photo facing north-west (Photo by Justus van Ramshorst).

The long term (1981-2010) average annual sum of precipitation is 662 mm and the average annual mean temperature is 9.6 °C; based on the Hanover weather station of the German Meteorological Service (station ID: 2014). Based on gap-filled meteorological data of our own grassland site in Mariensee, in 2020 and 2021 the annual precipitation was 521 mm and 597 mm, and the annual mean temperature was 11.3 °C and 9.8 °C, respectively. Based on gap-filled meteorological data of



Mariensee from 2019-2021, the long term mean wind speed at 3.0 m height was 1.87 m s$^{-1}$ and the dominant wind directions at the site were west and southwest.

The site was part of the "sustainable intensification of agriculture through agroforestry (SIGNAL) project", which investigates if and under which site conditions agroforestry can be a sustainable solution for future agriculture (Veldkamp et al., 2023). As part of the SIGNAL project, two EC towers were installed to measure and compare the micro-climate and CO$_2$
sequestration and evapotranspiration (ET) of the agroforestry grassland and the conventional grassland (Figure 1).

## 2.2    Instrumental setup

The grassland EC tower, was 3 m tall and placed west of the tree strips. The agroforestry EC tower was 10 m tall and placed next to the central tree strip. Both EC towers in Mariensee were equipped with similar instrumentation for meteorological measurements and EC (Table 1). Meteorological data were measured every 10 s and logged on a CR1000X data logger (Campbell
Scientific, Inc., Logan, UT, USA). The EC data, including an ultrasonic anemometer, were measured at a 2 Hz (LC-EC) or 20 Hz (CON-EC) frequency and logged on a CR6 data logger (Campbell Scientific, Inc., Logan, UT, USA).

### 2.2.1    Lower-cost eddy covariance

The LC-EC setups were present from the summer of 2019 until January 2022, however in the current study only data measured during the EC measurement campaigns in 2020 and 2021 was used for comparison. The LC-EC setup in the current study
was very similar to the ones used by Cunliffe et al. (2022). The LC-EC uses an uSONIC-3 Omni 3D ultrasonic anemometer (METEK GmbH, Elmshorn, Germany) and a closed-path gas analyser enclosure. Inside the custom made enclosure, the CO$_2$ mole fraction ($CO_2^{LC}$) was measured with a GMP343 IRGA (Vaisala Oyj, Helsinki, Finland) and inside the same cell the relative humidity ($RH_{LC}$) was measured with a HIH-4000 RH sensor (Honeywell International Inc., Charlotte, North Carolina, USA). The sensor response time of the GMP343 and HIH-4000 are 1.34 s and 4 s, respectively (Hill et al., 2017). The cell
temperature ($T_{CELL}^{LC}$) was measured using a fine wire thermocouple (Omega Engineering Inc., Norwalk, Connecticut, USA) with a 5 Hz response time. The absolute cell pressure ($P_{CELL}^{LC}$) was measured using a MPX5100AP pressure sensor (NXP USA Inc., Austin, Texas, USA). The enclosure consists of a heater, which can reduce the relative humidity inside the measuring cell during humid conditions, to prevent condensation. The vertical separation between the ultrasonic anemometer and the intake of the sampling tube was -0.2 m and the East- and Northward separation was 0 m. The Synflex 1300 tube (1300-M0603, Eaton
corporation, Dublin, Ireland) had a length of either 2 m (grassland) or 9 m (agroforestry) and an internal diameter of 4.0 mm and was fitted with two stainless steel 2 $\mu$m filters (SS-4FW-2, Swagelog, Solon, Ohio, USA). A nominal flow rate of $\sim$ 2 L min$^{-1}$ was achieved with a NMP830KNDC-B diaphragm gas pump (KNF Neuberger Inc., Trenton, New Jersey, USA). The flow rate could drop down to $\sim$ 1 L min$^{-1}$ when highly clogged. This resulted in a laminar flow with a Reynolds number of 717-358 inside the tubing (Massman, 1991).



**Table 1.** Meteorological and eddy covariance instruments, with height, model and company installed at both EC towers. All meteorological sensors were sampled every 10 s, except for precipitation which is the cumulative sum over 10 s. All EC sensors were either sampled at 2 Hz or 20 Hz.

| Variable | Height (m) | Model | Company |
|---|---|---|---|
| **Meteorological measurements** | | | |
| Net radiation, $R_N$ (W m$^{-2}$) | 2.5, 9.5 | NR-Lite2 | Kipp & Zonen, Delft, The Netherlands |
| Global radiation (downward and upward), $R_{G\downarrow}$, $R_{G\uparrow}$ (W m$^{-2}$) | 2.5, 9.5 | CMP3 pyranometer (2x) | Kipp & Zonen, Delft, The Netherlands |
| Relative humidity, $RH$ (%) and air temperature, $T$ (°C) | 2 | Hygro-thermo transmitter-compact (Model 1.1005.54.160) | Thies Clima, Göttingen, Germany |
| Precipitation, $P$ (mm) | 1 | Precipitation transmitter (Model 5.4032.35.007) | Thies Clima, Göttingen, Germany |
| Atmospheric pressure (only AF), $Pa$ (kPa) | 1 | Baro transmitter (Model 3.1157.10.000) | Thies Clima, Göttingen, Germany |
| Ground heat flux, $G_1$ and $G_2$ (W m$^{-2}$) | -0.05 | Hukseflux HFP01 (2x) | Hukseflux, Delft, The Netherlands |
| **EC measurements** | | | |
| 3D wind components, $u, v, w$ (m s$^{-1}$), and ultrasonic temperature, $T_s$ (°C) | 3, 10 | uSONIC-3 Omni | METEK GmbH, Elmshorn, Germany |
| Carbon dioxide mixing ratio, $CO_2$ (μmol mol$^{-1}$) | 3, 10 | LI-7200 | LI-COR Inc., Lincoln, NE, USA |
| Water vapour mixing ratio, $H_2O$ (mmol mol$^{-1}$) | 3, 10 | LI-7200 | LI-COR Inc., Lincoln, NE, USA |
| Carbon dioxide mixing ratio, $CO_2^{LC}$ (μmol mol$^{-1}$) | 3, 10 | GMP343 | Vaisala Oyj, Helsinki, Finland |
| Relative humidity, $RH_{LC}$ (%) | 3, 10 | HIH-4000 | Honeywell International Inc., Charlotte, North Carolina, USA |

### 2.2.2 Conventional eddy covariance

During three measurement campaigns in 2020 and 2021 CON-EC setups were installed and added to the existing LC-EC towers. The first campaign was at the grassland from the 3rd of June until the 25th of October 2020, the second at the agroforestry grassland from the 20th of August until the 26th of September 2020 and the third at the grassland from the 21st of July until the 26th of October 2021. The CON-EC setup shared the same uSONIC-3 Omni 3D ultrasonic anemometer (METEK GmbH,



Elmshorn, Germany) as the LC-EC. The $CO_2$ (µmol mol$^{-1}$) and $H_2O$ (mmol mol$^{-1}$) mixing ratios were measured using a LI-7200 enclosed-path infrared gas analyser (IRGA) (LI-COR Inc., Lincoln, NE, USA). The sensor response time of the LI-7200 for $H_2O$ was approximately $0.6 \pm 0.3$ s (Markwitz and Siebicke, 2019) and 0.16 s for $CO_2$. The vertical separation between the ultrasonic anemometer and the intake of the sampling tube was -0.2 m and the East- and Northward separation was 0 m. The insulated - but not heated - intake tube had a length of 1 m and an inner diameter of 8.2 mm. The flow rate was set at 15 L

min$^{-1}$, which results in a turbulent flow with a Reynolds number of 2623 inside the tubing (Leuning and King, 1992).

## 2.3 Flux processing

### 2.3.1 Lower-cost eddy covariance

**Pre-processing**

The LC-EC method requires some pre-processing steps before the eddy covariance calculations can be applied:

1. The LC cell pressure was smoothed using a 5-min centered moving average window in order to prevent additional noise being added to the covariance calculations.

2. The $H_2O^{LC}$ (mmol mol$^{-1}$) was calculated from the measured $RH_{LC}$, $T_{CELL}^{LC}$ and $P_{CELL}^{LC}$, following Markwitz and Siebicke (2019).

3. The mixing ratio $H_2O_{DRY}^{LC}$ (mmol mol$^{-1}$) was calculated following Burba et al. (2012).

4. The measured raw $CO_2^{LC}$ (µmol mol$^{-1}$, LC-EC uncorr.) mole fraction needed to be corrected for a variable cell temperature, relative humidity and pressure. This was not done automatically, only a variable cell temperature was used and constant values of pressure and relative humidity were assumed (LC-EC auto.). The final mixing ratio $CO2_{DRY}^{LC}$ (µmol mol$^{-1}$, LC-EC final) was calculated following the iterative equations provided by Vaisala (2023). The $CO_2$ correction required simultaneously measured $RH_{LC}$, $T_{CELL}^{LC}$ and $P_{CELL}^{LC}$, and several sensor specific temperature constants, which

could be pulled from each individual sensor memory.

5. The time lags of the LC-EC systems in the current study were considerably larger and more variable compared to a CON-EC setup with a LI-7200. This led to unsatisfactory time lag optimization when the standard time lag estimation method in EddyPro was applied. Therefore, realistic time lag windows for $CO_2$ and $H_2O$ were pre-estimated as follow in order to obtain an accurate time lag optimization in EddyPro. Based on the absolute maximum cross-correlation between the vertical wind speed ($w$) and $CO2_{DRY}^{LC}$, the time lag for $CO_2$ was estimated for each 30 minute data set. The nominal

time lag ($\tau^{nom}$) for each three measurement campaigns was estimated by determining the density peak of all 30-min time lags. The minimum ($\tau^{min}$) and maximum ($\tau^{max}$) time lag for each data set was calculated by multiplying the nominal time lag by 0.75 and 1.5, respectively (Table 2). The time lag window for $H_2O_{DRY}^{LC}$ was determined differently, as the time lag of $H_2O$ was more variable due to the effect of absorption and desorption of water. Nevertheless, it was





expected that the time lag of $H_2O$ was at least equal or longer than the time lag of $CO_2$. In order to avoid a too narrow window for the time lag optimization in EddyPro, $\tau_{H_2O}^{max}$ was fixed at 40 s for all three campaigns and $\tau_{H_2O}^{min}$ was assumed equal to $\tau_{CO_2}^{min}$.

**Table 2.** Estimated time lag windows for $CO_2$ during each measurement campaign.

|  | Grassland 2020 | Agroforestry 2020 | Grassland 2021 |
|---|---|---|---|
| $\tau_{CO_2}^{min}$ (s) | 4.83 | 6.29 | 5.20 |
| $\tau_{CO_2}^{max}$ (s) | 9.66 | 12.57 | 10.41 |

**Processing**

     The LC-EC fluxes based on the GMP343 and HIH-4000 were calculated using EddyPro (Version 7.0.3). The $CO_{2DRY}^{LC}$ and

$H_2O_{DRY}^{LC}$ were pre-calculated, as described in the previous paragraph. Also, meteorological data (air temperature, atmospheric pressure, relative humidity and global radiation) measured at the Mariensee stations were provided to EddyPro. During flux processing, double rotation, block averaging and automatic time lag optimization with predefined windows, as shown in the previous paragraph, were applied. The availability of mixing ratios made additional density (WPL) corrections redundant. Statistical tests for raw data screening were performed following Vickers and Mahrt (1997) and the random uncertainty estima-

tion due to sampling errors was calculated following Mann and Lenschow (1994). Corrections for spectral attenuation in the low-frequency range were performed following Moncrieff et al. (1997). High-frequency spectral attenuations were corrected following two methods, of which Horst (1997) was the main correction used in the current study. Additionally, spectral corrections following Ibrom et al. (2007), including Horst and Lenschow (2009) for sensor separation, were applied to investigate the sensitivity of the spectral correction method applied. Due to noisy spectra in the high-frequency range (see section 3.2.4),

the transfer function for the high-frequency correction was fitted from 0 Hz to 0.25 Hz.

### 2.3.2   Conventional eddy covariance

     The EC fluxes from the CON-EC setup were calculated using EddyPro (Version 7.0.3), and the applied flux processing was kept as similar as possible to the method applied for the LC-EC, in order to prevent additional uncertainties. The LI-7200 provides $T_{CELL}$ and $P_{CELL}$ measurements and instantaneous mixing ratios of $CO_2$ ($CO_{2DRY}$ (μmol mol$^{-1}$)) and $H_2O$

($H_2O_{DRY}$ (mmol mol$^{-1}$)), following Burba et al. (2012). The same meteorological data as for the LC-EC were provided to EddyPro. During flux processing, double rotation, block averaging and automatic time lag optimization (without predefined windows) were applied. Similar to the LC-EC calculations, the availability of dry mixing ratios made additional density (WPL) corrections redundant. Statistical tests for raw data screening were performed following Vickers and Mahrt (1997) and the random uncertainty estimation due to sampling errors was calculated following Mann and Lenschow (1994). Corrections for





spectral attenuation in the low-frequency range were performed following Moncrieff et al. (1997). High-frequency spectral attenuations were corrected following two methods, of which Horst (1997) was the main correction used in the current study. Additionally, spectral corrections following Ibrom et al. (2007), including Horst and Lenschow (2009) for sensor separation, were applied to investigate the sensitivity of the spectral correction method applied.

### 2.3.3    Quality control

For the $CO_2$, latent heat ($LE$) and sensible heat ($H$) fluxes from the CON-EC and LC-EC similar quality control (QC) was applied. Only the high quality data (Flag = 0) was used in the current study, based on the 0-1-2 flagging system according to Mauder et al. (2013). Fixed $u^*$ filtering was applied to the $CO_2$ and LE fluxes, similar to Cunliffe et al. (2022). For the grassland site the $u^*_{threshold}$ was set at 0.1 (m s$^{-1}$) and for the agroforestry site the $u^*_{threshold}$ was set at 0.15 (m s$^{-1}$). Furthermore, absolute limits for the $CO_2$, $LE$ and $H$ fluxes were applied. $CO_2$ fluxes below -30 μmol m$^{-2}$ s$^{-1}$ and above 30

μmol m$^{-2}$ s$^{-1}$ were discarded. $LE$ and $H$ fluxes below -50 W m$^{-2}$ and above 500 W m$^{-2}$ were discarded. After applying the combined QC, 54, 69 and 51% of the EC $CO_2$ fluxes were removed, and 52, 67 and 51% of the LC-EC $CO_2$ fluxes were removed, during the Grassland 2020, Agroforestry 2020 and Grassland 2021 campaign, respectively. For the EC $LE$ fluxes 62, 77 and 64% was removed, and 58, 74 and 59% of the LC-EC $LE$ fluxes were removed, during the Grassland 2020, Agroforestry 2020 and Grassland 2021 campaign, respectively. During nighttime, defined as incoming shortwave radiation <

20 W m$^{-2}$, more EC data were discarded than during daytime due to unfavorable turbulent conditions (Papale et al., 2006). For the three LC-EC campaigns combined this was also clearly visible, as 42% of the daytime data and 81% of the nighttime data were discarded based on the QC conditions. As the focus of this study was on instrument performance, we did not apply any gap-filling so that only measured data were compared.

### 2.3.4    Energy balance closure

The energy balance closure (EBC) for each EC system was assessed as an additional indicator for data quality. In the current study we used the energy balance closure as described in Equation 1, similar to Mauder and Foken (2006) and Reitz et al. (2022).

$$H + LE = R_N - G \qquad (1)$$

     With similar net radiation ($R_N$) and ground heat flux ($G$) for the CON-EC and LC-EC method, the difference between the

setups was caused by the sensible heat flux ($H$) and latent heat flux ($LE$) measured by the EC and LC-EC. Hence, even though the same ultrasonic anemometer was used for the EC and LC-EC setup, H was slightly different due to the humidity correction applied, which includes measurements of $ET$ (van Dijk et al., 2004). $G$ was the average of the two heat flux plates present, $G_1$ and $G_2$, when both were available. In the current study, soil and canopy storage were not measured and not included in the energy balance closure. However these storage terms would be the same for the EC and LC-EC method.





Additionally, the cumulative energy balance ratio (EBR) was also calculated and defined as the ratio of the total cumulative sum of the turbulent fluxes ($H + LE$) to the total cumulative sum of the available energy ($R_N - G$) (Cunliffe et al., 2022).

### 2.3.5 Statistical methods

Linear regression were calculated by applying a major axis regression with R package *lmodel2* (Legendre and Oksanen, 2018). The root mean square errors (RMSE) were calculated using R package *Metrics* (Hammer et al., 2018). The significance t-tests
were calculated using the R package *stats*.

## 3 Results

### 3.1 Meteorological conditions

In 2020 the annual mean air temperature was 1.7 °C above the long term average of 9.6 °C and the annual sum of precipitation was 21% below the long term average of 662 mm. In 2021 the annual mean air temperature was 0.2 °C above the long term
average and the annual sum of precipitation was 10% below the long term average. During the measurement campaigns, the mean RH and vapour pressure deficit (VPD) was 78.6% and 450 Pa and 83.8% and 299 Pa in 2020 and 2021, respectively. These results show that the campaign in 2020 was held during warmer and drier conditions compared to the campaign in 2021 (Figure 2). Additionally, the mean Bowen ratio during both campaigns also indicate that the conditions during the campaign in 2020 were less water abundant compared to 2021, as the mean Bowen ratio was 0.34 and 0.24 in 2020 and 2021, respectively.
Furthermore, during the measurement campaign in 2020 it was less windy compared to the campaign in 2021, with mean wind speeds of 1.38 m s$^{-1}$ and 1.54 m s$^{-1}$, respectively. Also during the campaign in 2020 it was more sunny compared to the campaign in 2021, as the mean incoming global radiation per day was 14.6 MJ m$^{-2}$ and 11.6 MJ m$^{-2}$, respectively. Finally, the average friction velocity, $u^*$, was higher during the agroforestry campaign compared to the grassland campaign in 2020, 0.33 m s$^{-1}$ versus 0.20 m s$^{-1}$.

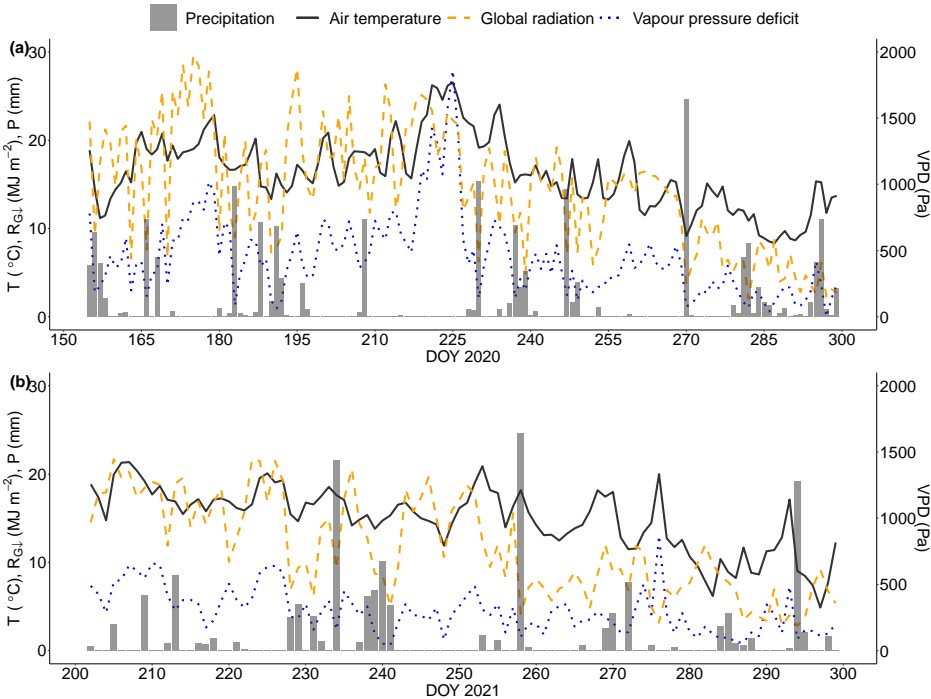

**Figure 2.** The meteorological conditions during the campaigns in 2020 and 2021. Daily mean values of the air temperature, T (°C) and vapour pressure deficit, VPD (Pa) are shown. Also, daily sums of precipitation, P (mm) and incoming global radiation, $R_{G\downarrow}$ (MJ m$^{-2}$) are shown.

## 3.2 Lower-cost versus conventional eddy covariance

### 3.2.1 Diurnal cycle

The diurnal pattern was clearly captured for the $CO_2$ and LE fluxes by both EC setups and during all campaigns, with $CO_2$ uptake and water vapor release during the day and $CO_2$ release and dew fall during night (Figure 3). The negative $CO_2$ fluxes during midday of the LC-EC were lower relative to the CON-EC during all campaigns. The strong positive $CO_2$ fluxes of the the LC-EC were only lower relative to the CON-EC in 2020. The mean of the average diurnal $CO_2$ cycle for both EC setups was positive during both grassland campaigns, 1.06 µmol m$^{-2}$ s$^{-1}$ in 2020 and 0.87 µmol m$^{-2}$ s$^{-1}$ in 2021, and was negative during the agroforestry campaign, -0.65 µmol m$^{-2}$ s$^{-1}$. The diurnal pattern of the LE flux was very similar for both EC setups during the grassland campaign in 2020, nevertheless during nighttime the EC setups agree less and the diurnal cycle was more noisy. For example, the LE flux of the CON-EC at the agroforestry site was higher compared to the LE flux of the LC-EC during the first 6-7 hours of the day, however this coincides with time periods when limited amount of data was available. The LE flux at the grassland site in 2021 has a similar diurnal pattern between EC setups, however the magnitudes were different





and opposite to the 2020 campaigns, as in 2021 the daytime LE flux of the LC-EC has a higher magnitude compared to the CON-EC.

The diurnal pattern of the sensible heat flux (H) was also captured and shows a very strong agreement between the LC-EC
and CON-EC, which share the same ultrasonic anemometer (figure not shown in the current paper). Nevertheless, the LC-EC has a slightly higher H compared to the CON-EC during midday, reflecting slight differences in the humidity correction for H, and this difference was larger for the grassland sites.

### 3.2.2 Scatter plots

$CO_2$ and LE fluxes of the LC-EC and CON-EC were strongly correlated with $r = 0.95\text{-}0.98$ and $r = 0.92\text{-}0.95$ for the $CO_2$ and
LE fluxes, respectively (Figure 4). Furthermore, the linear regression results in slopes between 0.87 and 0.93 ($R^2 = 0.91\text{-}0.95$) for the $CO_2$ fluxes, and slopes between 1.02 and 1.23 ($R^2 = 0.84\text{-}0.9$) for the LE fluxes. The LC-EC $CO_2$ fluxes were generally lower than the CON-EC $CO_2$ fluxes, indicated by the slopes from linear regression between 0.87 and 0.93. The agreement for $CO_2$ fluxes between both EC setups was better for positive fluxes (slope = 0.97-1.12, $R^2 = 0.54\text{-}0.75$) than for negative fluxes (slope = 0.77-0.93, $R^2 = 0.68\text{-}0.81$). The correlation between the LE fluxes of both EC setups was lower compared
to the $CO_2$ fluxes, especially for the grassland sites, which was also visible by the relatively large spread that increases with higher LE fluxes. Nevertheless, the slopes for the grassland and agroforestry campaigns in 2020 were very good, 1.02 and 1.03, respectively. However in 2021, the slope between the LE fluxes at the grassland site was 1.23 ($R^2 = 0.84$), indicating that the LE flux of the LC-EC setup was 23% higher compared to the CON-EC setup. The distribution of the positive LE fluxes in 2021 looks very similar to the LE fluxes in 2020, however the magnitude of the LE fluxes does not agree. Furthermore, the negative
LE fluxes disagree even more, which indicates differences between EC setups during humid conditions.

The scatter plots of H show a very strong correlation between the LC-EC and EC setup, with a $r = 1.0$, which corresponds with the use of the same ultrasonic anemometer (figures not shown). The H fluxes measured with the LC-EC setups were slightly higher compared to the H fluxes measured with the EC setups, due to humidity effect corrections which include measurements of ET, resulting in a slope of 1.02 ($R^2 = 1.0$) for the grassland campaigns and a slope of 1.01 ($R^2 = 1.0$) for the
agroforestry campaign.



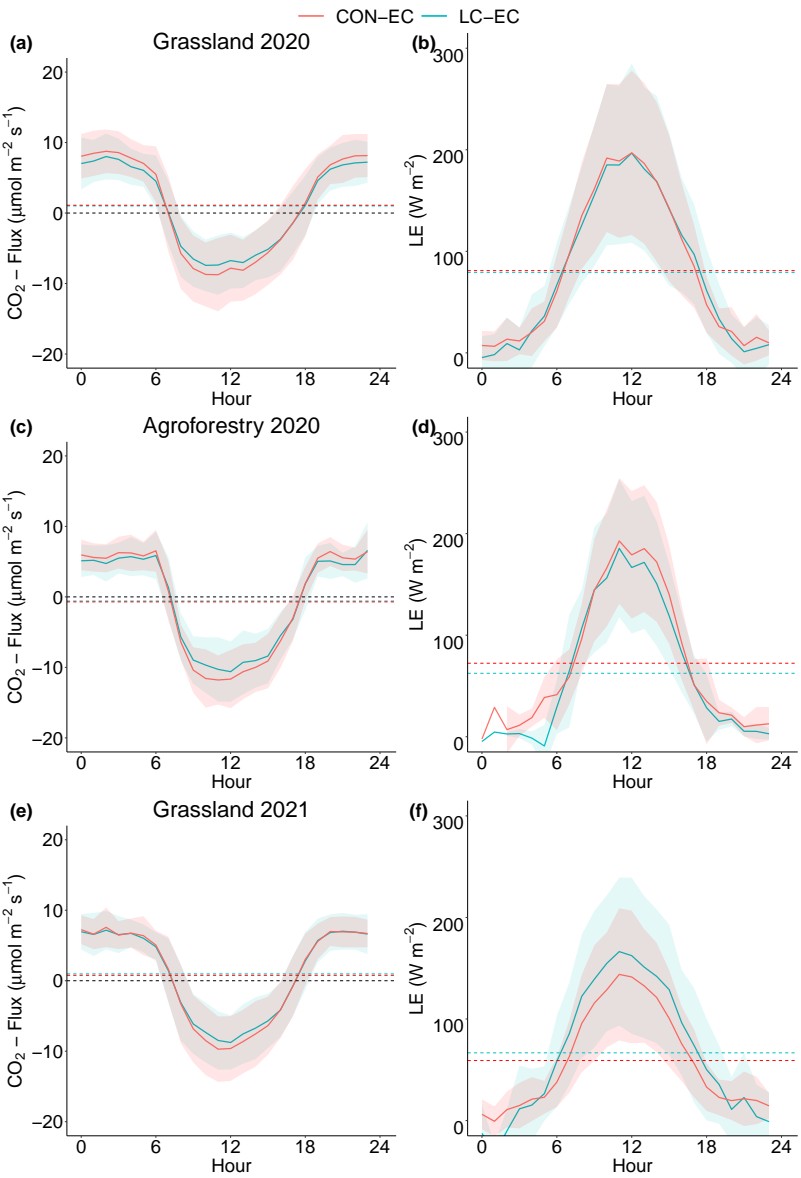

**Figure 3.** Mean diel cycles of $CO_2$ and LE fluxes (mean $\pm$ standard deviation) based on the entire campaign, measured with the CON-EC (red) and the LC-EC (light blue) setup for the grassland site in 2020 (a) and (b), the agroforestry site in 2020 (c) and (d) and the grassland site in 2021 (e) and (f). The black dashed lines in the figures of the $CO_2$ flux highlight when the flux is zero and the flux changes sign. A negative flux indicates $CO_2$ is sequestered and a positive flux indicates $CO_2$ emitted. The red and light blue dashed lines indicate the mean of each diel cycle of the CON-EC and LC-EC, respectively.

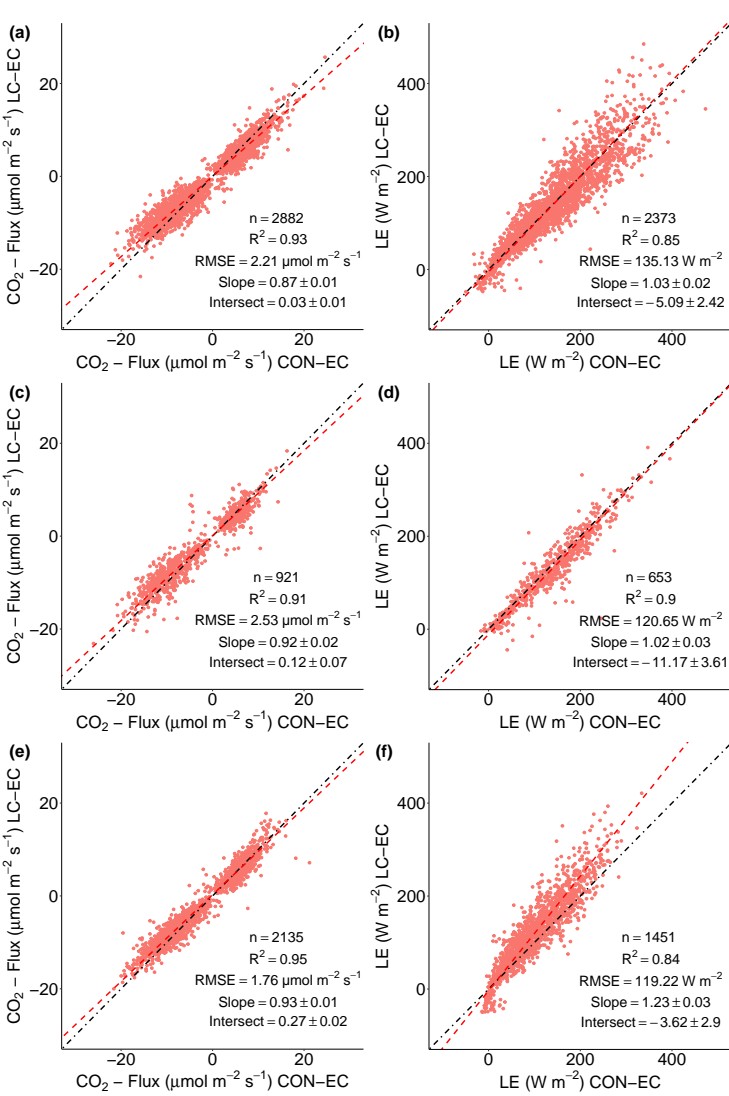

**Figure 4.** Half-hourly $CO_2$ and LE fluxes measured with LC-EC versus half-hourly $CO_2$ and LE fluxes measured with CON-EC for the grassland site in 2020 (a) and (b), the agroforestry site in 2020 (c) and (d) and the grassland site in 2021 (e) and (f).



### 3.2.3 Energy balance closure

The energy balance closure (EBC) at the grassland site in 2020 was similar for both EC setups, with a slope of 0.84 ($R^2$ = 0.9 and 0.81), however the CON-EC has a higher correlation with the available energy compared to LC-EC, $r = 0.95$ versus $r = 0.9$, respectively (left column in Figure 5). The agroforestry site in 2020 shows a very high EBC for both EC setups, with

a slope of 1.0 ($R^2$ = 0.83) and 0.98 ($R^2$ = 0.85) for the LC-EC and CON-EC, respectively. The correlation at the agroforestry site was more similar for the LC-EC and CON-EC, with $r = 0.91$ versus $r = 0.92$, respectively. The EBC at the grassland site in 2021 shows the biggest difference between EC setups. A slope of 0.83 ($R^2$ = 0.83) from the LC-EC was similar compared to 2020. In contrary, the EBC of the CON-EC has a lower slope of 0.75 ($R^2$ = 0.93), despite the high correlation of $r = 0.96$.

The cumulative energy balance ratio (EBR) at the grassland site in 2020 was very similar for both EC setups, with a total

closure ratio of 93.5% and 92.7% for the LC-EC and CON-EC, respectively (right column in Figure 5). The agroforestry site in 2020 shows also a very high EBR closure ratio for both EC setups, 97.5% and 100.7% for the LC-EC and CON-EC, respectively. The EBR also shows the biggest difference between EC setups at the grassland site in 2021. An EBR closure ratio of 91.4% from the LC-EC was similar compared to 2020. In contrary, an EBR closure ratio of 78.9% from the CON-EC was different compared to 2020.


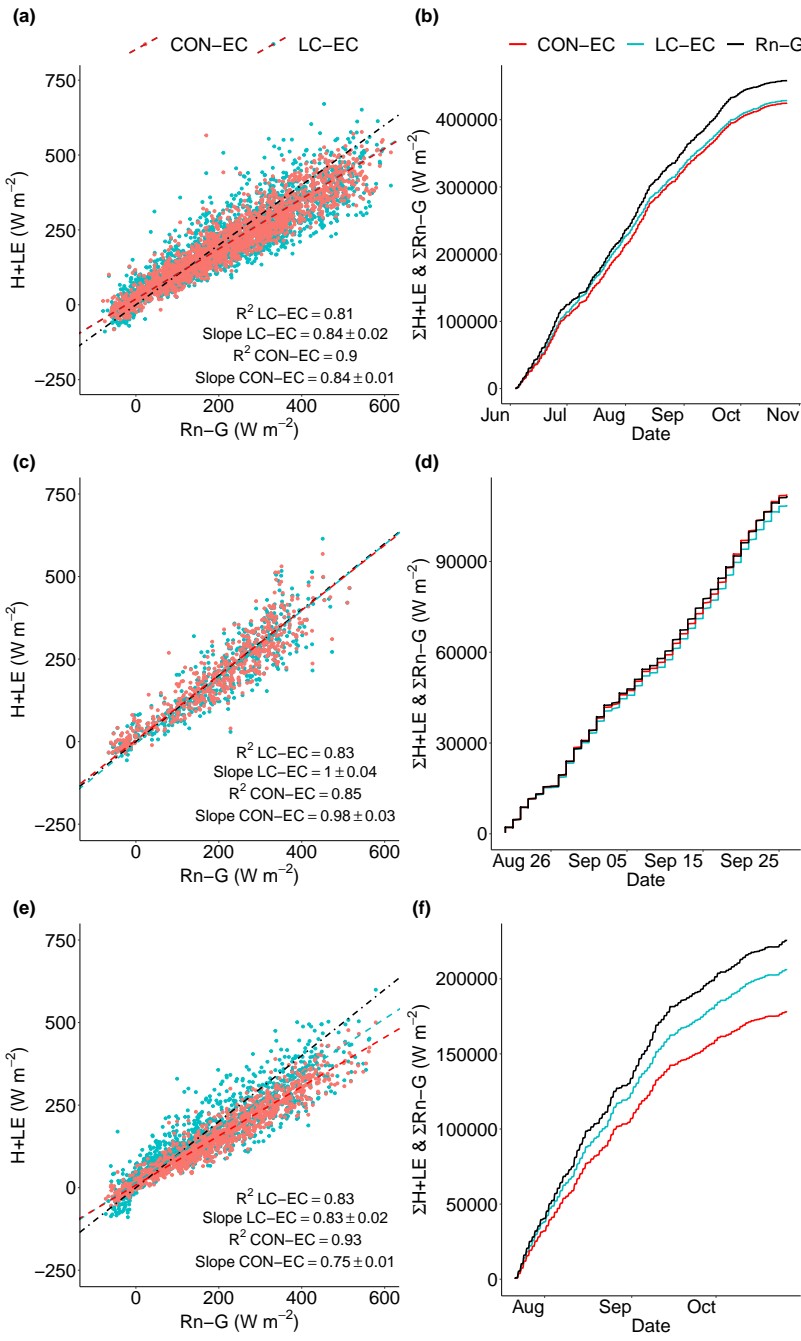

**Figure 5.** The energy balance closure (EBC) with half-hourly turbulent fluxes ($H + LE$) measured with the CON-EC (red) and the LC-EC (lightblue) setup, versus the available energy ($R_N - G$). The EBC is shown for the grassland site in 2020 (a), the agroforestry site in 2020 (c) and the grassland site in 2021 (e). The cumulative energy balance ratio (EBR) is showing the cumulative sum of the half hourly turbulent fluxes measured with the CON-EC (red) and the LC-EC (light blue) setup, and the cumulative sum of the available energy (black). The cumulative EBR is shown for the grassland site in 2020 (b), the agroforestry site in 2020 (d) and the grassland site in 2021 (f).





### 3.2.4 Spectral analysis

In general the spectra of the LC-EC show a stronger decay in energy content compared to the spectra of the CON-EC in the higher frequency range, which was a consequence of the slower sensor response time of the LC-EC sensors (Figure 6). Furthermore, for both EC setups the $H_2O$ spectra always show more attenuation compared to the $CO_2$ spectra and the loss was increased during higher RH conditions, as visualized for RH-classes of 50% and 80% (Figure 6b, d, e). However, the $H_2O$ spectra of the heated LC-EC were less affected by the RH conditions compared to the non-heated CON-EC, and the taller AF tower seems less affected by the RH conditions compared to the short grassland towers as well.

All the spectra of the CON-EC show the effect of aliasing of high-frequency signal, clearly visible at the frequencies just under the Nyquist frequency of 10 Hz, where the energy content of the power spectra increases in energy due to folding of unresolved signal of frequencies higher than the Nyquist frequency of the CON-EC (Stull, 1988; Massman, 2000). At the same time the effect of (random white) noise seems to be apparent in the $CO_2$ and $H_2O$ spectra as well, expressed by the spectral energy increasing all the way up to a slope of +1. The effect of noise was increasingly present at the $H_2O$ spectra during higher RH conditions. The LC-EC shows a similar effect of aliasing for the $T$ spectra at frequencies just below 1 Hz, the Nyquist frequency of the LC-EC.

Due to a logging issue at both campaigns in 2020, the ultrasonic anemometer of the CON-EC was sampled at 2 Hz while logged at 20 Hz, which continuously resulted in ten repetitive values. This was due to oversampling, which is visible by the harmonic oscillations in the high frequencies of the $T$ spectra, starting at 1 Hz (Eugster and Plüss, 2010). As the LC-EC was measuring at 2 Hz, the effect of oversampling was not present in the LC-EC $T$ spectra. The $CO_2$ and $H_2O$ spectra of the LC-EC were also affected by oversampling, as the frequency response time of the $CO_2$ and $H_2O$ sensors is lower than the 2 Hz measurement rate. Based on the frequency response times found by Hill et al. (2017), the oversampling rate can be approximated for the $CO_2$ and $H_2O$ sensors as follows, $2/0.74 = 2.7$ and $2/0.25 = 8$, respectively. The oscillations were clearly visible in both spectra, however the shape of the spectra and oscillations look differently. The $CO_2$ spectra of the LC-EC shows a similar harmonic oscillation as the $T$ spectra of the CON-EC, but additionally there is an increased spectral energy at lower frequencies due to aliasing. Different from the $CO_2$ spectra, the $H_2O$ spectra of the LC-EC were affected by random white noise, which results in a loss of sensor signal, visualized by the slope of +1 (Figure 6). As there is no signal distinguishable from the high amount of noise, there is unresolved signal to fold back, hence the seemingly unaffected shape of the spectra left of the $H_2O$ sensor's Nyquist frequency. The lack of signal also leads to peaks in the $H_2O$ spectra instead of harmonic oscillations as seen in the $CO_2$ spectra (Eugster and Plüss, 2010).

The cospectra of the LC-EC also show a stronger decay compared to the spectra of the CON-EC in the higher frequency range, again a consequence of the slower sensor response time of the LC-EC sensors (Figure 7). Furthermore, the $Co(wC_{H_2O})$ cospectra for both EC setups show more decay compared to the $Co(wC_{CO_2})$ cospectra. The LC-EC $Co(wC_{CO_2})$ and $Co(wC_{H_2O})$ cospectra have a higher spectral energy in the lower frequencies compared to the CON-EC due to aliasing of higher frequencies. Moreover, the LC-EC $Co(wC_{CO_2})$ and $Co(wC_{H_2O})$ cospectra were quite similar for each setup, however the higher AF tower seems the least affected and the grassland tower in 2021 seems the most affected.





All the cospectra of both EC setups show an increase in spectral energy at the higher end of the frequencies, which seems
to be an consequence of the noise sources described in the spectra, namely random white noise, aliasing and oversampling.
However, clearly some cospectra were affected earlier by the noise than others, and the harmonic oscillations of the spectra
were not visible in the cospectra. The $Co(wT)$ cospectra of the CON-EC in 2020 were more affected and increase in spectral
energy earlier than the $Co(wT)$ cospectra of the CON-EC in 2021, which seems to be a direct result of the wrong sampling
frequency of the ultrasonic anemometer. Also the $Co(wC_{CO_2})$ and $Co(wC_{H_2O})$ cospectra of the CON-EC in 2021 appear
less affected compared to the 2020 cospectra. The $Co(wT)$ cospectra of the LC-EC follow a similar shape compared to the
CON-EC $Co(wT)$ cospectra, and were the best at the higher AF tower and slightly worse at the grassland towers.

### 3.2.5 Correction of CO$_2$ concentration

The automatic correction by Vaisala (LC-EC auto.), which only considers a variable cell temperature ($T_{CELL}^{LC}$) and assumes
constant values of pressure and relative humidity, improved the CO$_2$ mixing ratio compared to the raw CO$_2$ mole fraction
(LC-EC uncor.) (Figure 8). Nevertheless, it is clearly visible that when the full correction was applied (LC-EC final), also
considering a variable cell pressure ($P_{CELL}^{LC}$) and cell relative humidity ($RH_{LC}$), the CO$_2$ mixing ratio was closest to the
CO$_2$ mixing ratio measured by the LI-7200 (CON-EC). The LC-EC auto. correction increases the mean CO$_2$ concentration
compared to the LC-EC uncor. by 3-4% and the LC-EC final decreases the mean CO$_2$ concentration compared to the LC-EC
uncor. by 2-3%. For the agroforestry 2020 and grassland 2021 campaign the offset between the LC-EC and EC is relatively
constant during the day. For the grassland 2020 campaign the offset between the LC-EC and EC is not constant and larger
during midday.



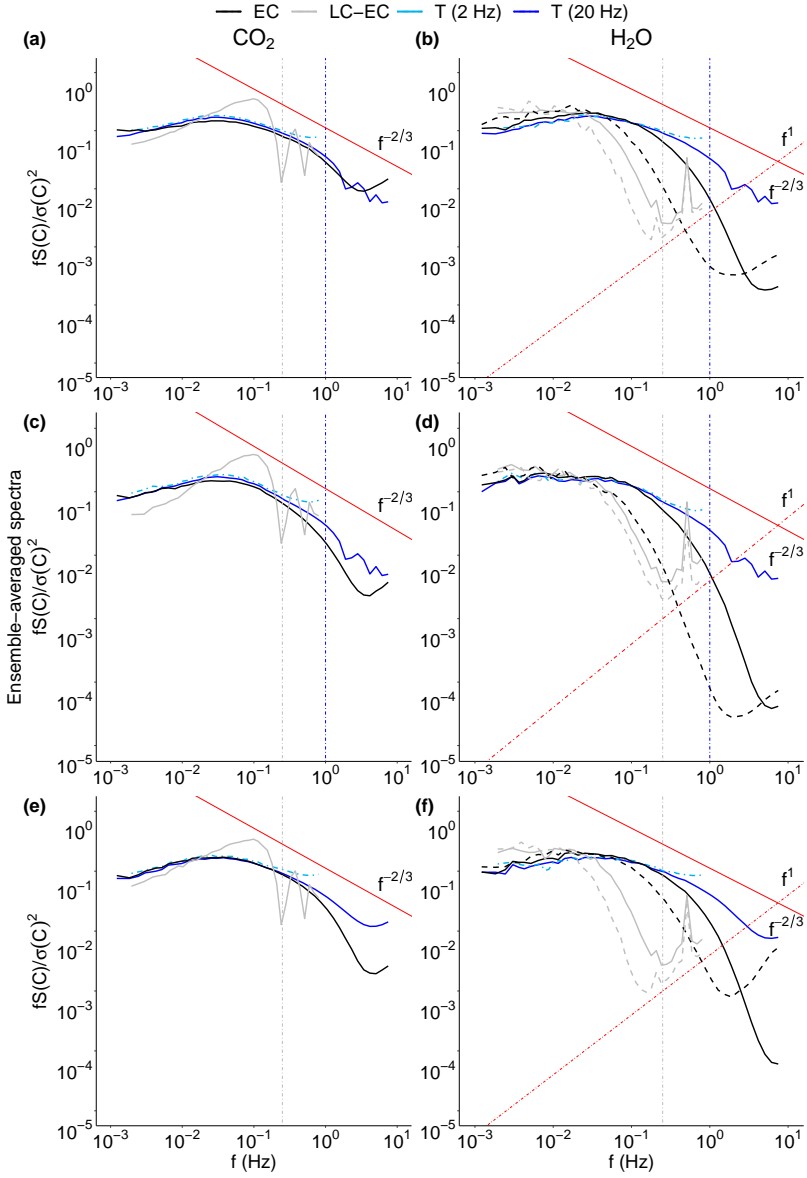

**Figure 6.** Ensemble-averaged normalized $CO_2$ (left column), $H_2O$ (right column) and $T$ spectra versus the natural frequency ($f$). The $CO_2$ and $H_2O$ spectra of the LC-EC setup (grey) and the CON-EC setup (black) are shown, and also the $T$ spectra of the LC-EC setup (dash-dotted light blue) and the CON-EC setup (blue) are shown. The $H_2O$ spectra are shown for relative humidity bins of 45-55% (solid lines) and 75-85% (dashed lines). The spectra for the grassland site in 2020, agroforestry site in 2020 and grassland site in 2021 are shown in subfigure (a) and (b), (c) and (d), and (e) and (f), respectively. The grey dash-dotted lines at 0.25 Hz are to visualize the fitting range for the high-frequency correction of the LC-EC. The blue dash-dotted lines at 1 Hz are to visualize the oversampling of the ultrasonic anemometer data in 2020. The solid red lines with a -2/3 slope indicate the theoretical decay of the spectra in the inertial subrange and the dash-dotted red lines with a +1 slope indicate the slope for random white noise.

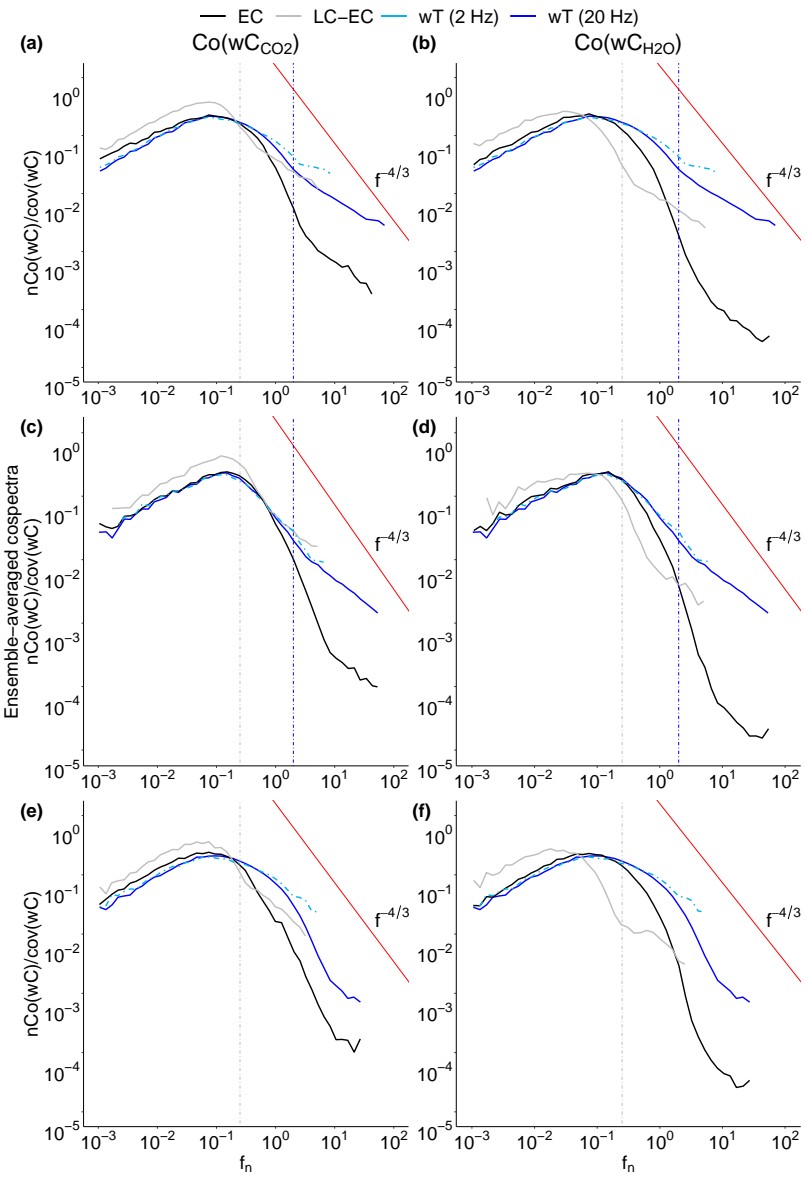

**Figure 7.** Ensemble-averaged normalized $Co(wC_{CO_2})$ (left column), $Co(wC_{H_2O})$ (right column) and $Co(wT)$ cospectra versus the normalized frequency ($f_n$) for unstable conditions. The $Co(wC_{CO_2})$ and $Co(wC_{H_2O})$ cospectra of the LC-EC setup (grey) and the CON-EC setup (black) are shown, and also the $Co(wT)$ cospectra of the LC-EC setup (dash-dotted light blue) and the CON-EC setup (blue) are shown. The cospectra for the grassland site in 2020, agroforestry site in 2020 and grassland site in 2021 are shown in subfigure (a) and (b), (c) and (d), and (e) and (f), respectively. The grey dash-dotted lines at 0.25 Hz are to visualize the fitting range for the high-frequency correction of the LC-EC. The blue dash-dotted lines at 2 Hz are to visualize the oversampling of the ultrasonic anemometer data in 2020. The solid red lines with a -4/3 slope indicate the theoretical decay of the cospectra in the inertial subrange.



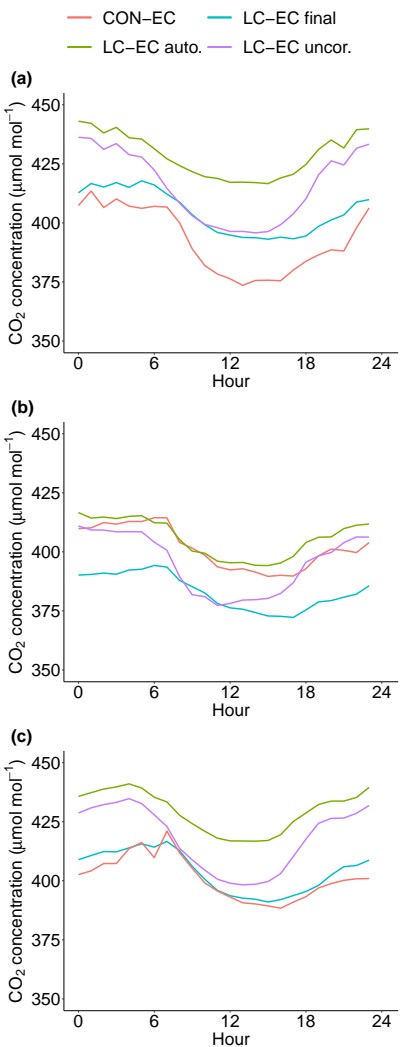

**Figure 8.** Average diurnal cycles of $CO_2$ concentrations based on the entire campaign. Four different $CO_2$ concentrations are shown, (i) the $CO_2$ mixing ratio measured with a LICOR LI-7200 (CON-EC (red)), (ii) the final corrected $CO_2$ mixing ratio measured with the LC-EC (LC-EC final (blue)), (iii) the raw $CO_2$ mole fraction measured by the LC-EC (LC-EC uncor. (purple)) and (iv) the automatic Vaisala corrected $CO_2$ mixing ratio measured with the LC-EC (LC-EC auto. (green)). The average diurnal cycles are shown for the grassland site in 2020 (a), the agroforestry site in 2020 (b) and the grassland site in 2021 (c).





## 3.3 Effect of the spectral correction method on cumulative fluxes

The cumulative $CO_2$ and $ET$ fluxes show a variety of differences across the spectral correction methods of Horst (1997) and Ibrom et al. (2007), which can be summarized by three observations from Figure 9: (i) The difference between spectral correction methods for the cumulative $CO_2$ fluxes was varying between 0.04-12.5%, which was lower compared to the differences between the cumulative $ET$ fluxes, which were varying between 5.63-38.8%. (ii) The differences between spectral correction methods at the agroforestry site were 0.04% and 5.63-16.4% for the cumulative $CO_2$ and $ET$ fluxes, respectively. This was lower compared to the differences between spectral correction methods at the grassland sites, which were 2.44-12.5% and 8.43-38.8% for the cumulative $CO_2$ and $ET$ fluxes, respectively. (iii) The differences between the spectral correction methods for the cumulative $CO_2$ and $ET$ fluxes from the CON-EC setups were varying between 0.04-9.85%, which was lower compared to the 0.04-38.8% difference between the cumulative $CO_2$ and $ET$ fluxes from the LC-EC setups. The spectral correction factors (SCF's) of each setup show that these three observations correlate with the magnitude of the SCF (Figure 10). The higher the SCF, the higher the relative difference between spectral correction methods. Furthermore, the SCF was always higher for the Horst method compared to the Ibrom method (Figure 10). Accordingly, the Horst method leads to a higher closure of the energy balance, compared to the Ibrom method, 78.9-100.7% versus 64.6-96.9%, respectively. (Table 3).

The $ET$ flux of the grassland campaign in 2021 was different compared to the 2020 campaigns for two reasons (Figure 9 f). (i) The difference between spectral correction methods at the LC-EC setup was 16.5% higher in 2021 compared to the same grassland in 2020. (ii) The CON-EC SCF's in 2021 were lower and show less spread compared to both campaigns in 2020 (Figure 10). As a consequence of the lower SCF's, the energy balance ratio at the grassland in 2021 was only 74.1-78.9%, compared to 86.1-92.7% in 2020 (Table 3). Finally, the $CO_2$ flux of the CON-EC in 2021 looks reasonable and has a slightly higher SCF compared to 2020.

**Table 3.** Energy balance ratios (EBR) of the three measurement campaigns and for two different spectral correction methods, Horst (1997) and Ibrom et al. (2007).

|  | Grassland 2020 | Agroforestry 2020 | Grassland 2021 |
|---|---|---|---|
| EC (Horst) (%) | 92.7 | 100.7 | 78.9 |
| LC-EC (Horst) (%) | 93.5 | 97.5 | 91.4 |
| EC (Ibrom) (%) | 86.1 | 96.9 | 74.1 |
| LC-EC (Ibrom) (%) | 71.6 | 86.4 | 64.6 |

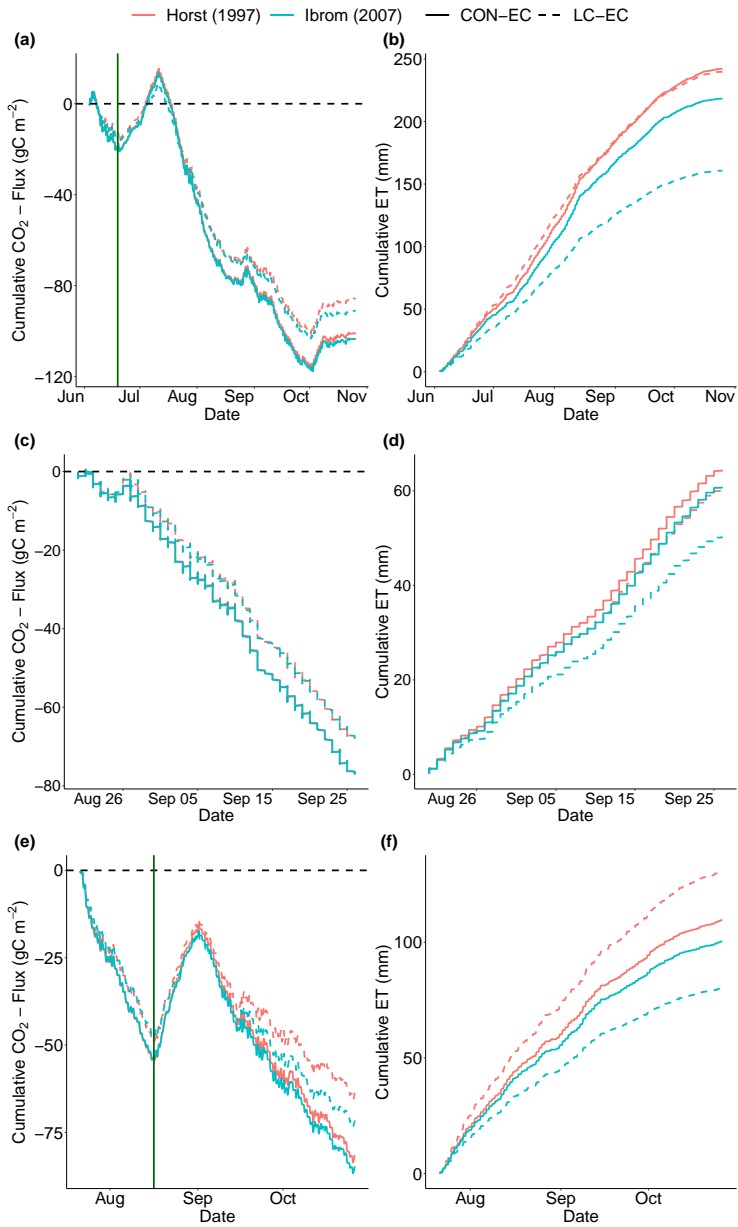

**Figure 9.** Non gap-filled cumulative $CO_2$ (left column) and $ET$ (right column) fluxes of the three measurement campaigns and for two different spectral correction methods, Horst (1997) and Ibrom et al. (2007). The grassland site in 2020 is shown in (a) and (b), the agroforestry site in 2020 is shown in (c) and (d), and the grassland site in 2021 is shown in (e) and (f). The red lines are cumulative fluxes processed with the Horst method and and the light blue lines are cumulative fluxes processed with the Ibrom method. The solid lines are the CON-EC fluxes and the dashed lines are the LC-EC fluxes. The vertical solid green lines in (a) and (e) indicate when the grassland was mowed. The horizontal black dashed lines in (a), (c) and (e) indicate the transition of the ecosystem being either a $CO_2$ source (+) or sink (-).



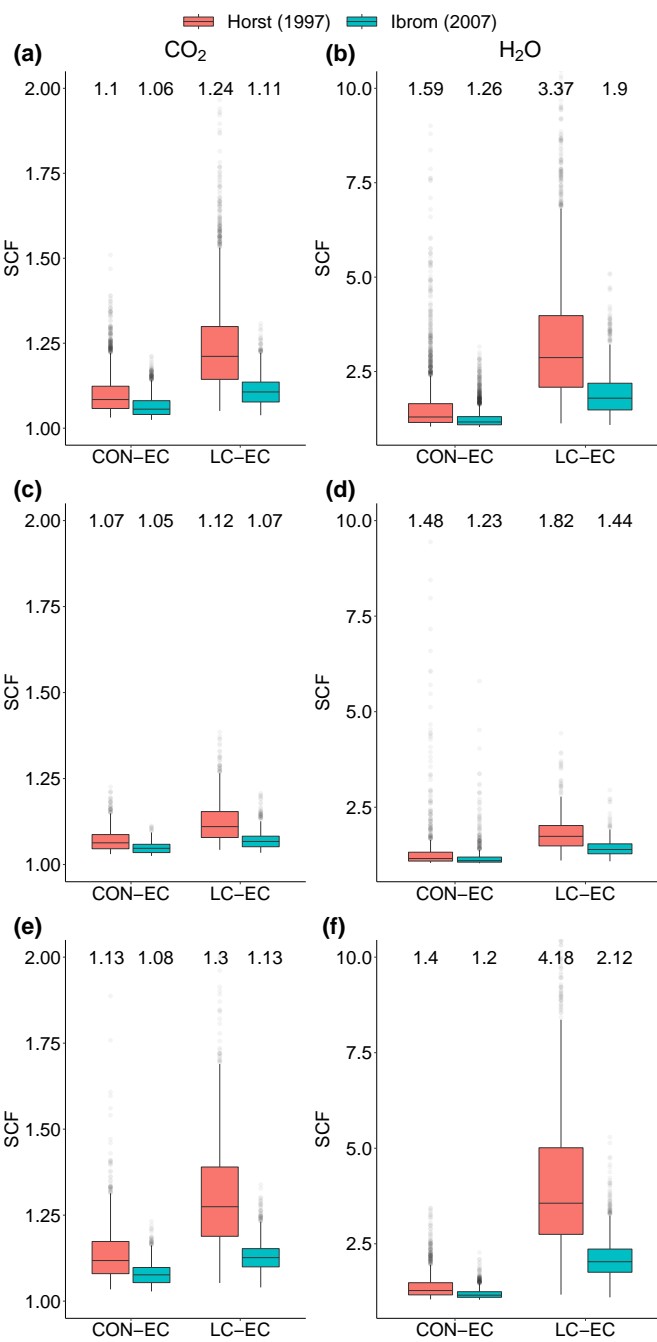

**Figure 10.** Boxplots of the $CO_2$ (left column) and $H_2O$ (right column) spectral correction factors (SCF's) of the three measurement campaigns and for two different spectral correction methods, Horst (1997) and Ibrom et al. (2007). The grassland site in 2020 is shown in (a) and (b), the agroforestry site in 2020 is shown in (c) and (d), and the grassland site in 2021 is shown in (e) and (f). The red boxes are the SCF's of the Horst method and the lightblue boxes are SCF's of the Ibrom method, and are shown for both EC setups separately. For the boxplots only the SCF's of the quality controlled data are used. The value above each boxplots indicates the mean SCF.





## 3.4 Ecological application

**Cumulative fluxes**

Both EC setups well capture the temporal variability of $CO_2$ fluxes such as diel pattern (Figure 3) as well as mowing events,
e.g. at 19th June 2020 and 16th August 2021 (Figure 9). Both EC setups also capture temporal variability of $ET$ showing that
$ET$ decreases towards the end of the growing season.

Even though the cumulative fluxes of the LC-EC and EC agree quite well, the magnitude of the cumulative fluxes show
a difference between EC setups varying between 1.01-28.0% for the Horst method (Table 4), which was an aggregation of
structural offsets between the $CO_2$ and $ET$ flux measured by the LC-EC and CON-EC during parts of the day (Figure 3). For
the $ET$ measurements the difference between EC setups was higher with the Ibrom method than with the Horst method (Table
4). In contrary, for the $CO_2$ fluxes the difference between EC setups was equal or higher for the Horst method than with the
Ibrom method.

**Table 4.** The relative differences of the cumulative $CO_2$ and $ET$ fluxes, between the LC-EC and EC setups and between the Horst (1997)
and Ibrom et al. (2007) spectral correction methods. The relative differences were calculated based on the final value of the cumulative sums
of $CO_2$ and $ET$ or each EC setup and spectral correction method.

| Difference in % | Grassland 2020 | Agroforestry 2020 | Grassland 2021 |
|---|---|---|---|
| $CO_2$ : LC-EC ($\frac{Horst-Ibrom}{Horst}$) | 6.33 | 0.04 | 12.5 |
| EC ($\frac{Horst-Ibrom}{Horst}$) | 2.44 | 0.04 | 4.01 |
| Horst ($\frac{\text{LC-EC}-EC}{\text{LC-EC}}$) | 18.0 | 13.1 | 28.0 |
| Ibrom ($\frac{\text{LC-EC}-EC}{\text{LC-EC}}$) | 13.7 | 13.2 | 18.3 |
| $ET$ : LC-EC ($\frac{Horst-Ibrom}{Horst}$) | 33.0 | 16.4 | 38.8 |
| EC ($\frac{Horst-Ibrom}{Horst}$) | 9.85 | 5.63 | 8.43 |
| Horst ($\frac{\text{LC-EC}-EC}{\text{LC-EC}}$) | 1.01 | 7.11 | 16.4 |
| Ibrom ($\frac{\text{LC-EC}-EC}{\text{LC-EC}}$) | 35.9 | 21.0 | 25.1 |

**Agroforestry versus grassland**

In 2020 the grassland and agroforestry sites were measured simultaneously for about one month, and in Figure 11 the not
gap-filled cumulative $CO_2$ and $ET$ flux for this period were compared. During this month, the average cumulative $CO_2$ se-
questration of both EC setups was about 2.3 times higher at the agroforestry site compared to the grassland site, -70.3 versus
-30.4 g C m$^{-2}$, respectively ($p < 0.001$ for LC-EC and $p < 0.001$ for CON-EC). The average cumulative $ET$ of both EC setups
was similar at the agroforestry and grassland site, 52.1 and 51.9 mm, respectively ($p > 0.05$). The difference between LC-EC
and conventional EC setups was similar for both ecosystems ($p > 0.05$ for $CO_2$ and ET), and the difference in cumulative sums





between agroforestry and grassland was also roughly similar for both EC setups ($p > 0.05$ for $CO_2$ and ET). Additionally, for the $CO_2$ flux the EC setup difference was smaller than the ecosystem difference. Finally, the agroforestry sites has a higher efficiency in using water as the $CO_2$ flux$/ET$ ratio was 2.3 times higher compared to the grassland.

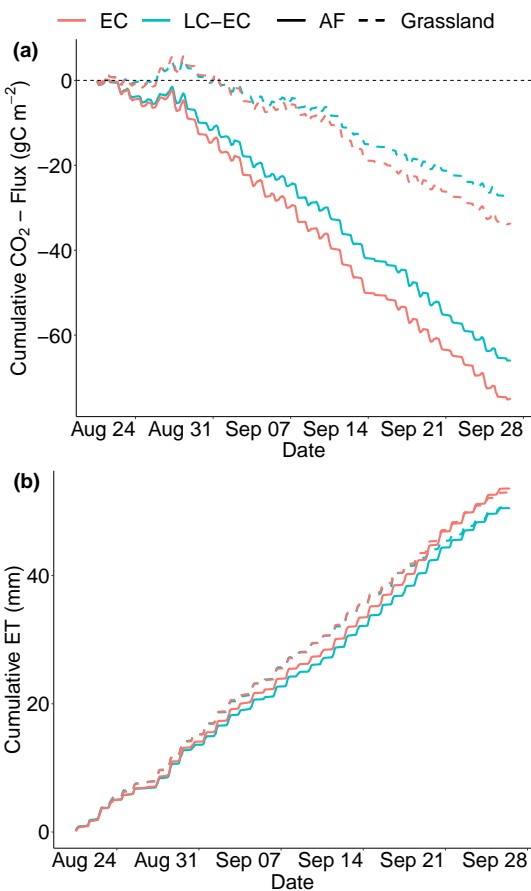

**Figure 11.** Non gap-filled cumulative $CO_2$ (a) and $ET$ (b) fluxes of the agroforestry and grassland site during the period they were measured simultaneously in 2020. The red lines are the CON-EC fluxes and the lightblue lines are the LC-EC fluxes. The dashed lines are the grassland site and the solid lines are the agroforestry site. The horizontal black dashed line in (a) indicates the transition of the ecosystem being either a $CO_2$ source (+) or sink (-).





## 4 Discussion

### 4.1 Technical characterisation

The current study showed that the LC-EC was also able to capture the diel pattern and ecosystem response of the $CO_2$ and LE
fluxes observed at the grassland and agroforestry grassland by the CON-EC. The LC-EC setup showed a strong correlation with
the CON-EC, with $r$ = 0.95-0.98 and $r$ = 0.92-0.95 for the $CO_2$ and LE fluxes, respectively (Figure 4). The LC-EC $CO_2$ flux
was lower compared to CON-EC, indicated by the linear regression slopes of 0.87-0.93 ($R^2$ = 0.91-0.95), which can be clearly
observed from the average diel cycles in Figure 3. The LC-EC LE fluxes in 2020 were close to CON-EC, indicated by linear
regression slopes of 1.02 and 1.03 ($R^2$ = 0.85-0.9), and have similar diel cycles. The LE fluxes in 2021 did not agree well. This
observation will be discussed elaborately in section 4.1. Furthermore, the energy balance closure (EBC) and cumulative energy
balance ratio (EBR) of both EC setups agreed very well in 2020, which confirms the reliability of capturing the energy fluxes
of both EC setups (Figure 5). Finally, the (co)spectra of both EC setups showed a very clear difference, as a consequence of
the slower response time of the LC-EC $CO_2$ and $H_2O$ sensors (Figure 6 & 7). This lead to consistent higher spectral correction
factors for the LC-EC setup compared with CON-EC (Figure 10).

**Comparison to other lower-cost eddy covariance studies**

To put the results of the current study in perspective, a comparison is made with the few existing recent studies comparing $CO_2$
and $H_2O$ fluxes of a LC-EC setup and a CON-EC setup.

The study of Hill et al. (2017) compared a predecessor of the current LC-EC setup, which had a higher flow rate of approx-
imately 75 L min$^{-1}$, with an open-path LI-7500 IRGA at a 4.25 m tall tower on a pasture in Dumfries and Galloway, UK.
Despite the different CON-EC IRGA and a higher flowrate, their results agree quite well with the current study. Their $CO_2$
fluxes had a better agreement in magnitude, with a linear regression slope of 1.03 and 0.983 compared to 0.87-0.93, however
the coefficient of determination ($R^2$) between their EC setups was less with a $R^2$ of 0.86 and 0.72, compared to $R^2$ between
0.91 and 0.95. It has to be noted that the amount of QC in their study was minimal, which probably lead to lower $R^2$ as
compared to the extensive QC in the current study. The $H_2O$ fluxes of both studies were quite similar, with a linear regression
slope of 1.06 ($R^2$ = 0.89), compared to 1.02 ($R^2$ = 0.9) and 1.03 ($R^2$ = 0.85). Even with the turbulent conditions inside the
sampling tube and the higher flow rate, the average spectral correction factors (SCF's) of the $CO_2$ flux of Hill et al. (2017)
were higher compared to our study, 1.52-1.55 compared to 1.12-1.3. The SCF of the LE flux of Hill et al. (2017) was 2.33,
which was lower than the SCF of the grassland towers, 3.37 and 4.18, but higher than the SCF of 1.82 at the agroforestry tower.
Furthermore, they noted that the agreement of the LC-EC $CO_2$ flux with the LI-7500 got worse with lower magnitude $CO_2$
fluxes, which was probably a consequence of a lower signal-to-noise ratio.

The study of Cunliffe et al. (2022) used the exact same LC-EC enclosure as the current study, at a 6.0 m tall tower in the
northern Chihuahuan Desert, USA. The fluxes were compared with a LI-7500, however the measurements do not take place
at one and the same tower, but at four nearby towers. Furthermore, the fluxes were affected by a low signal-to-noise ratio, due
to the low magnitude of fluxes in a dry desert ecosystem. For fluxes at a daily timescale, their LC-EC LE fluxes showed a





worse performance compared to CON-EC, with the LC-EC LE fluxes being approximately 6-22% lower, compared to LC-EC LE fluxes 2-3% higher for half-hourly fluxes. However, their cumulative ET - including gap-filling - looks similar to the ET measurements at the agroforestry tower of the current study. The $CO_2$ flux of Cunliffe et al. (2022) was severely affected by the low magnitude of $CO_2$ fluxes, which led to a low correlation between the LC-EC's and CON-EC setup, and LC-EC $CO_2$ fluxes

being lower with a slope of approximately 0.48 for fluxes at a daily timescale, compared to a slope of 0.87-0.93 for half-hourly fluxes. The clearly noisy $CO_2$ fluxes of Cunliffe et al. (2022) also result in a high uncertainty of the cumulative $CO_2$ fluxes.

The unpublished work by Callejas-Rodelas et al. (2023) used the exact same LC-EC enclosure as the current study, at a 3.5 m tall tower on a crop field in Wendhausen, Germany. The fluxes of the three LC-EC setups at one single tower were also compared with a LI-7200, however the flux calculations were performed using the EddyUH software (Mammarella et al.,

2016), and the high-frequency corrections were applied following the method from Mammarella et al. (2009). Their $CO_2$ fluxes across the LC-EC setups had a better agreement in magnitude with linear regression slopes between 0.95-1.05 compared to 0.87-0.93, but a similar high $R^2$ between the EC setups of 0.88-0.92 compared to 0.91-0.95. Their $H_2O$ fluxes across the LC-EC setups performed worse, with lower slopes between 0.78-0.99 compared to 1.02-1.03, but similar $R^2$ of 0.85 compared to 0.85-0.9 (LC-EC setup with issues excepted). As a consequence of the lower LE fluxes for both the LC-EC and CON-EC in

their study, the energy balance closure was worse compared to the current study, 66-74% compared to 83-84%. Moreover, the LI-7200 from the unpublished work by Callejas-Rodelas et al. (2023) potentially also underestimates the LE flux similar as in the current study, indicated by the low EBC and the big difference in ET compared to agroforestry (section 4.1).

For an even wider perspective, the study of Polonik et al. (2019) is very useful, comparing $CO_2$ and $H_2O$ fluxes of five types of conventional IRGA's and three types of ultrasonic anemometers on a 4 m tall tower at the edge of an alfalfa field in Davis,

California. Even though these were all conventional - high cost - EC setups, the spread of the linear regression slope between EC setups varied between 0.92 to 1.08 for $CO_2$ fluxes and 0.74 to 1.36 for $H_2O$ fluxes, depending on the spectral correction method. Hence, almost all the linear regression slopes of the $CO_2$ and $H_2O$ fluxes of the current study fit within this range, even though the tower of the current study was 1 m lower. Finally, in the current study we compared the LC-EC with a LI-7200, however the study of Polonik et al. (2019) highlights that there is no absolute truth, which means carefulness is needed when

comparing the performance of EC setups.

**Detailed technical characterisation**

The EBC of both EC setups during the two grassland campaigns in the current study fit within the observed range of $0.86 \pm 0.20$ for grasslands of the FLUXNET database (Stoy et al., 2013). Nevertheless, the EBC of the CON-EC in 2021 was lower and agreed better with the EBC of a wetland of $0.76 \pm 0.13$ (Stoy et al., 2013). The EBC of the agroforestry site was roughly

17.5% higher compared to the grassland sites, which can be explained by the more heterogeneous landscape, which results in increased turbulent conditions and a higher friction velocity, $u^*$, at the agroforestry tower (Franssen et al., 2010; Stoy et al., 2013). Moreover, not measuring the storage components (soil, air and biomass) of the energy balance at the agroforestry site, might give a biased image of the EBC, as the tree strips could potentially store energy.



When a EC tower is taller, the high-frequency eddies become less important and the cospectrum peak and the energy-
containing eddies shift to lower frequencies, and opposite, closer to the ground the higher frequency eddies are more important
(Moncrieff et al., 1997; Reitz et al., 2022). This effect was clearly seen when the high-frequency spectral correction factors of
the 10 m tall agroforestry tower were compared to the 3 m tall grassland tower (Figure 10). The SCF's for both $CO_2$ and $H_2O$
fluxes, and for both EC setups, were lower at the agroforestry site. When comparing the 2020 campaigns it becomes clear that
this effect was larger for the LC-EC than for the CON-EC, because at a tall tower it is less problematic that this setup is not
able to measure the high-frequency eddies, due to a higher occurrence of low-frequency eddies which seem to better fit the
slower response time of the $CO_2$ and RH sensor (Markwitz and Siebicke, 2019). Furthermore, it is important to note that the
high-frequency spectral correction (method) becomes also less important when the tower is higher, as there is less loss which
needs to be compensated (Mauder and Foken, 2006). To summarize, the performance of the LC-EC probably improves with
increasing tower height, however this must be possible within the targeted ecosystem, as the footprint size increases with tower
height.

One of the differences between the EC setups was the flow rate and the consequent laminar or turbulent flow regime inside
the inlet tube. Turbulent flow conditions inside the inlet tubes are generally preferred because the high-frequency attenuation
is less compared to laminar flow conditions (Leuning and Moncrieff, 1990; Suyker and Verma, 1993; Moncrieff et al., 1997).
Nevertheless, the tube attenuation can be characterised by the Reynolds number, and turbulent flow conditions do not per
definition lead to less attenuation compared to laminar flow conditions (Massman, 1991). Furthermore, a higher flow rate
requires more power and more cleaning maintenance due to the increase in pollutants inside the tubing and filters (Moncrieff
et al., 1997). Also, it needs to be considered that tube attenuation affects the higher frequencies, which are not measured by the
LC-EC setup anyway, due to the slow response of the $CO_2$ and $H_2O$ sensors. So higher turbulent flow rates might not reduce
the attenuation of the LC-EC that much, compared to CON-EC setups, as observed when the SCF's of the current study were
compared with the SCF's of Hill et al. (2017). Moreover, it was noteworthy that the agroforestry site, with a 9 m long tube,
has a lower attenuation than the grassland site, with a 2 m long tube, which shows that other design aspects as height might
be more important for the LC-EC setup (Leuning and Moncrieff, 1990). In general, a shorter tube length would likely reduce
the flux attenuation and the time lag, something which can be considered in future designs of the LC-EC setup (Leuning and
Moncrieff, 1990).

Finally, two considerations for future LC-EC studies: (i) a LC-EC design with shorter inlet tubes would probably reduce
attenuation. Additionally, the unpublished work by Callejas-Rodelas et al. (2023) suggests to also heat these shorter inlet
tubes, in addition to heating the enclosure, to prevent condensation and potential erroneous data. (ii) In the current study only
the highest quality data (flag = 0) was used, which for both EC setups led to discarding of 51-77% of the data, which is not
uncommon, especially at nighttime (Papale et al., 2006; Mauder et al., 2013). Nevertheless, for future long term ecosystem
flux analysis this would lead to large gaps and therefore using high and moderate quality data (flag = 0 and 1) is recommended.
This would increase the noise of the fluxes, however the unpublished work by Callejas-Rodelas et al. (2023) shows that the
correlation between the LC-EC and CON-EC was still good with such quality control, and instead 39-45% of the data was
discarded.





**Spectral characterisation**

The spectra and cospectra were already described in detail in section 3.2.4, however the distortions due to noise, aliasing and oversampling are discussed more elaborately in this section.

   The random white noise and aliasing effects were visible in all spectra and cospectra, however these do not affect the flux calculations. The random white noise is not correlated with the vertical wind speed and therefore makes no systematic contribution to the fluxes (Rummel et al., 2002). Aliasing is the folding of unresolved signal above the Nyquist frequency

into frequencies below the Nyquist frequency, which distorts the shape of the (co)spectra, but this does not influence the total flux calculations (Stull, 1988; Massman, 2000). Aliasing can occur because the Nyquist frequency is lower than the sensor response time (Stull, 1988), but aliasing in the low frequency range is also possible when the sensor is incorrectly representing the energy of the higher frequencies (Markwitz and Siebicke, 2019). The aliasing of the cospectra in the lower frequency range and an increase in spectral energy in the high-frequency range was also observed by the LC-EC setup of Markwitz and Siebicke

470    (2019).

   The effect of oversampling was clearly visible in the spectra. For the CON-EC setups in 2020, the oversampling of the $T$ spectra only led to small harmonic oscillations and as a consequence no strong aliasing (Eugster and Plüss, 2010). The $CO_2$ and $H_2O$ spectra of the LC-EC were more affected. The LC-EC $CO_2$ spectra was affected by a combination of oversampling and aliasing, something which is observed by Eugster and Plüss (2010) for high oversampling rates. The strong oscillations are

not uncommon, however the location of the aliasing was different than the standard aliasing just below the sampling Nyquist frequency, either 1 or 10 Hz, as described before. Based on the peaks of the oscillations it was possible to determine the sensor Nyquist frequency and the response time of the $CO_2$ sensor, as described by Eq. 1 in section 4.3 of Eugster and Plüss (2010). The first peak of the oscillations was at $\sim 0.37$ Hz, which can be converted into a sensor Nyquist frequency of $\sim 0.123$ Hz, and a sensor response time of $\sim 0.25$ Hz. A 4 s sensor response time fits the length of the complete measurement sequence

of the GMP343 $CO_2$ sensor, which is 4 s (Hill et al., 2017). Nevertheless, a single measurement of the GMP343 within the complete sequence lasts 1.36 s, and this was found to be the optimal time response for the frequency corrections by Hill et al. (2017) and the unpublished work by Callejas-Rodelas et al. (2023). The LC-EC $H_2O$ spectra were affected by a combination of oversampling and the absence of signal in the frequencies higher than $\sim 0.25$ Hz. The absence of signal leads to the observed peak at $\sim 0.5$ Hz in the spectra instead of oscillations (Eugster and Plüss, 2010). Furthermore, the $H_2O$ spectra confirms the

observed sensor response time of 0.25 Hz by Hill et al. (2017), as beyond this frequency no signal is distinguishable from noise.

**Underestimation of the latent heat flux in 2021**

   The general characterisation of the LC-EC and CON-EC fluxes were discussed in the previous section, however the $H_2O$ flux of the CON-EC in 2021 will be discussed in more depth since these results were poor.

First of all, it was not expected that the SCF for the LE flux from the CON-EC setup was lower in 2021 compared to 2020, as Fratini et al. (2012) shows that a higher RH and wind speed would probably lead to a higher SCF for the LE flux,



something which was not observed in the current study for both spectral correction methods (Figure 10). On the other hand, the studies of Barr et al. (1994) and Brotzge and Crawford (2003) measured and De Roo et al. (2018) modeled that the EBC decreased when the Bowen ratio decreased. As 2021 was wetter, colder and had a lower Bowen ratio compared to 2020, this

could explain that the LI-7200 performs worse in 2021. Additionally, Stoy et al. (2013) reports that wetlands, with likely more humid conditions and a lower Bowen ratio compared to less wet environments, on average have a lower EBC compared to normal grasslands. Recently, the study of Zhang et al. (2023) also showed the consistent underestimation of LE fluxes in the high quality FLUXNET2015 dataset, especially for closed and enclosed path sensors during high RH conditions above 70%. In the current study, 51% of the quality controlled LE data in 2021 has a RH inside the IRGA above 70%, compared to 31% in

2020, confirming that the data in 2021 was more likely affected by similar issues.

More specific to the LI-7200, the study of Metzger et al. (2016) suggests heating of the inlet, to prevent having RH levels inside the IRGA above 60%, which are considered problematic. In the current study, 77% and 54% of the quality controlled LE measurements in 2021 and 2020 consist of a RH level inside the IRGA higher than 60%, respectively. In retrospect, heating the LI-7200 could have prevented the issue visible with the LE data in 2021, as similarly the heated LC-EC enclosure does

not show this issue. Nevertheless, this is not a guarantee issues will not occur, as the study of Perez-Priego et al. (2017) used an insulated and heated inlet, but still reports strong underestimations of up to 35% of the LE flux using a LI-7200. Especially during humid and high RH conditions in the growing season these large errors occured and the underestimation was much larger at the shorter tower (1.5 m) compared to the tall tower (15 m) (Perez-Priego et al., 2017).

Based on the results presented it is not possible to point at a clear cause of the LE underestimations in 2021 and why this is

not happening in 2020. It is clear that the difference in LE and EBC between the CON-EC and LC-EC increases with higher RH in 2021 (data not shown), which confirms that the effect of water plays an important role in the EBC (Stoy et al., 2013). However, the same effect was not visible in 2020 during high RH conditions, which suggests that the magnitude of RH is not the only important element. Additionally, the study of Zhang et al. (2023) mentions the importance of spectral correction methods which take into account the effect of RH, but at the same time also notes that potentially also these do not fully correct for the

observed biases. The current study confirms that both the Horst (1997) and Ibrom et al. (2007) spectral correction methods lead to an underestimation of the LE flux in 2021. This suggests that the issue was independent of the spectral correction method, but could for example point at a transfer function which badly represents the actual attenuation. For example, De Ligne et al. (2010) and Emad (2023) argued that using a first order linear filter to fit the non-linear behavior of the $H_2O$ spectral attenuation, might not be the most accurate. Nevertheless, the linear IIR-fit obtained with EddyPro in 2021 was not perfect, but also not very

poor or worse than in 2020, which suggests that something else than the spectral correction might play a role in the observed underestimations of the latent heat flux (Figure A1).

## 4.2 Effect of the spectral correction method

The results showed that the relative effect of the spectral correction method on the flux magnitude increases with higher spectral correction factors, or in other words, with an increasing loss of high-frequency signal. When the relative importance of

the spectral correction method increases, systematic small differences between spectral correction methods are added up and





the difference between spectral methods and the total uncertainty of the flux increases (Mauder and Foken, 2006; Reitz et al., 2022). As the LC-EC per definition has stronger loss of high-frequency signal, applying the right spectral correction method is more important compared to CON-EC. Based on our results, and especially the better energy balance closure and energy balance ratio, the Horst (1997) method was chosen as preferred spectral correction method in the current study, even though

the Ibrom et al. (2007) method was designed for closed path EC setups. As the LC-EC fluxes where still deviating from the CON-EC, it would be interesting to test a wide variety of other spectral correction methods in the future, especially because the system design of the LC-EC is different from CON-EC setups which have been used and thoroughly tested in the past (Polonik et al., 2019; Reitz et al., 2022).

### 4.3 Ecological application

The LC-EC setup was able to measure the $CO_2$ and LE flux above the grassland and agroforestry grassland, including ecosystem disturbances as grass mowing. During simultaneous measurements at the agroforestry and grassland site, there was a significant difference in cumulative carbon uptake. Despite the significant difference in carbon uptake, some caution is needed regarding the magnitude of the difference between ecosystems, as there was only one month of non gap-filled data and more night than daytime data was discarded. Nevertheless, it was likely that the agroforestry site sequesters more carbon, as the

recent study by Veldkamp et al. (2023) which includes the grassland site of the current study, showed that there was a significant difference in carbon sequestration between agroforestry and monoculture grasslands. Furthermore, trees on agricultural land globally contribute significantly to carbon uptake and storage (Zomer et al., 2016). During the same period there was no significant difference in cumulative ET, similar to what was observed by Markwitz et al. (2020) at several agroforestry sites in Germany.

### 5 Conclusions


The current study showed at an agroforestry and grassland site in a temperate ecosystem that lower-cost eddy covariance (LC-EC) can be a cheaper alternative for the costly conventional EC (CON-EC). There was a strong correlation between the $CO_2$ and latent heat flux measurements of the closed-path LC-EC and the CON-EC with an enclosed-path LI-7200. The LC-EC $CO_2$ fluxes were slightly lower than the CON-EC, and the LE flux was equal for both EC setups in the 2020. In 2021 the LE flux of

the LC-EC was of similar quality as in 2020, however the LE flux of the CON-EC seemed to be affected by underestimations.

The (co)spectra of the LC-EC were more attenuated in the high-frequency range compared to the CON-EC due to the slower response sensors of the LC-EC setup. Both EC setups were affected by random white noise and aliasing in the spectra, and in addition the $CO_2$ and $H_2O$ LC-EC spectra were affected by oversampling. The high-frequency spectral corrections for the LC-EC were higher compared to the CON-EC, but this difference could be reduced by taller towers, when the ecosystem

footprint is not violated, as the cospectrum shifts to lower frequencies. The difference between spectral correction methods increased with higher spectral corrections, and therefore the spectral correction had an increased effect on the LC-EC fluxes, particularly for the more attenuated $H_2O$ flux. Both EC setups measured a significantly higher cumulative carbon uptake at





the agroforestry site compared to the grassland site, and an equal cumulative ET for both ecosystems during one month of simultaneous measurements.

Finally, the results show that LC-EC has the potential to measure EC fluxes at various land-use systems for approximately 25% of the costs of a CON-EC system. LC-EC setups can be used to increase the spatial representativeness of flux measurements in heterogeneous ecosystems. Design-wise a shorter and heated inlet tube would be recommended and future in-depth investigations in an optimal spectral correction method could lead to further optimization of the spectral corrections.

*Data availability.* Dataset with fluxes calculated using EddyPro will be made available through a DOI when the paper is accepted.

*Author contributions.* JGVvR performed the measurements and data analysis and wrote the manuscript. AK contributed to data analysis and manuscript writing and wrote the project proposal. JACR contributed to data analysis and manuscript writing. RC contributed to data analysis and manuscript writing. TCH contributed to data analysis and manuscript writing. CM contributed to data analysis, manuscript writing and wrote the project proposal.

*Competing interests.* The authors declare that they have no conflict of interest.

*Acknowledgements.* The authors are thankful for the fruitful discussions with Anas Emad and other colleagues at the Bioclimatology group; we also acknowledge the technical support through field work received from Marek Peksa and Dietmar Fellert (Bioclimatology group) and Dirk Böttger and Julian Meyer (Soil Science group of Tropical and Subtropical Ecosystems) from the University of Göttingen.

*Financial support.* This research has been supported by the German Federal Ministry of Education and Research (BMBF; project BonaRes, Modul A: SIGNAL; grant no: 031B0510A). We also acknowledge the support by the Open Access Publication Funds of the University of
Göttingen.



## Appendix A: Comparison of linear IIR-fit at grassland sites in 2020 and 2021

**Figure A1.** The ratio of ensemble-averaged normalized $\frac{H_2O}{T}$ spectra (solid line) of the CON-EC versus the natural frequency. Additionally, the linear IIR-fit obtained with EddyPro (dashed line), which represents the transfer function for the high-frequency corrections used for the CON-EC $H_2O$ flux calculations. The ratios and transfer functions are shown for the 2020 (grey) and 2021 (black) grassland campaign and presented in five RH-classes bins obtained with Eddypro.



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
