# Peer review of "Lower-cost eddy covariance for CO2 and H2O fluxes over grassland and agroforestry"

_Atmospheric Measurement Techniques, 2024_

## Author Comment (AC1)

*7th of August 2024*

This document contains a point by point reply to review #1 and #2 of "Lower-cost eddy covariance for $CO_2$ and $H_2O$ fluxes over grassland and agroforestry" by Justus van Ramshorst et al. (DOI: https://doi.org/10.5194/amt-2024-30).

In this document you can find:

- The replies to review #1 from the 7th of April 2024 at page 2 (https://doi.org/10.5194/amt-2024-30-RC1)
- The replies to review #2 from the 17th of June 2024 at page 15 (https://doi.org/10.5194/amt-2024-30-RC2)

**Review #1**

**Summary**

This manuscript presents comparisons between lower cost and conventional eddy covariance instrumentation at two landscapes – a grassland and an agroforestry site in Germany. The work shows compelling evidence that the lower cost system can be successful at measuring CO2 and H2O fluxes in these landscapes. The work is of interest to the readership of AMT and is of generally good quality. However, I think it can be improved significantly through more hypothesis driven objectives, clearer writing, and more concise presentation. Some logging errors (a sonic at 2Hz instead of 20Hz) should be more adequately addressed. I have major, minor, and technical comments that should be considered. I rate each of its scientific significance, scientific quality, and presentation quality as "Good".

*Thank you for your kind words and constructive comments. Below you can find a point by point reply to your review.*

**Major comments**

1. We don't learn until L279 that in 2020 the CON-EC system is sampled at 2 Hz but logged at 20 Hz, so the values are repeated ten times, which causes harmonic oscillations. This needs to be addressed in the methods and more explicitly considered. I'd tend to just report the fluxes and cospectra from 2Hz and not 20, but maybe there are other ways to do this (e.g. focusing on spectra from the gas analyzers). Even consider really emphasizing the 2021 dataset through the paper and then using 2020 for a more supportive role.

*We fully understand this comment and similar the comment by reviewer #2, who suggested to reprocess the CON-EC data from 2020 in 2 Hz instead of 20 Hz. Following your suggestions, we now more explicitly mentioned the logging error in the method section and emphasized that the CON-EC data in 2020 is in 2 Hz (Section 2.2 and 2.2.2, L110).*

*All the 2020 fluxes and spectra are recalculated and this (mostly) improved the agreement (slopes) between the LC-EC and CON-EC in 2020, as visible below in the scatterplots. Furthermore, we updated all figures, results and discussion using the 2 Hz CON-EC fluxes/results in 2020.*

[Figure]

2. There are many graphs – are they all needed?

*Reviewer #2 suggested to move Figure 8 to the supplement, so we did this accordingly.*

3. Fig 11 is not gap-filled and loses a lot of value that way. Can a simple gap-filling approach (e.g. with REDDYPROC) be used to assess the impact of these site and instrument differences in a real use-case?

*We agree that gap-filling this figure adds value and therefore gap-filled the data for the figure. The agroforestry datasets are unfortunately too short for REddyProc (< 3 months), but instead we used the XGboost gap-filling method by Vekuri et al. 2023. A description of this additional processing step is added to method section 2.3.4. and reads as follows:*

As the focus of this study was on instrument performance, we did not apply any gap-filling when comparing the LC-EC and CON-EC setups, so that only measured data were compared. Therefore, Figures 1–9, A1 and B1 include quality controlled, but non gap-filled data. As an exception, Figure 10 uses gap-filled data to illustrate a real use-case of comparing cumulative ecosystem fluxes of an agroforestry and grassland system. For the gap-filling high and moderate quality data (Flag = 0 or 1) were selected (Mauder et al., 2013). Subsequently, the gap-filling was done using XGBoost with five predictors: air temperature, vapour pressure deficit (VPD), global radiation, wind speed and wind direction (Vekuri et al., 2023). The gap-filling uncertainty was evaluated by calculating the standard deviation (SD) of the bias distribution between the measured and modeled 30-min fluxes, and propagated through the cumulative sum by multiplying $2 \cdot \text{SD}$ with the squared root of the number of 30-min filled gaps.

*The gap-filling changes the results of the simultaneous cumulative fluxes, as shown below, therefore section 3.4 – Agroforestry versus grassland now includes the updated Figure 10 (previously 11) and is rewritten as follows:*

**Agroforestry versus grassland**

In 2020 the grassland and agroforestry sites were measured simultaneously for about one month, and in Figure 10 the gap-filled cumulative $CO_2$ and $ET$ flux for this period were compared. During this month, the agroforestry site was a carbon sink of -67.9 g C m$^{-2}$ and the grassland site a carbon source of 37.7 g C m$^{-2}$, based on the average cumulative $CO_2$ sequestration of both EC setups ($p < 0.001$ for LC-EC and CON-EC). The $CO_2$ flux difference between EC setups was smaller than the ecosystem difference, 19.6 g C m$^{-2}$ and 8.1 g C m$^{-2}$ for the grassland and agroforestry site, respectively. Similarly, the average gap-filling uncertainty for both EC setups was also smaller than the ecosystem difference, 3.2 g C m$^{-2}$ and 3.6 g C m$^{-2}$ for the grassland and agroforestry site, respectively. The cumulative $ET$ of both EC setups shows a less clear message, the $ET$ was higher for both EC setups at the grassland site than at the agroforestry site, however the CON-EC was 14.4 mm higher ($p < 0.001$) and the LC-EC was 3.1 mm higher ($p > 0.05$). The average gap-filling uncertainty for both EC setups was 1.5 mm and 1.4 mm for the grassland and agroforestry site, respectively. Furthermore, for the $CO_2$ and $ET$ fluxes the difference between the LC-EC and CON-EC was large at the grassland site ($p < 0.001$ for $CO_2$ and $ET$). The difference in cumulative sums between agroforestry and grassland was smaller with the LC-EC setup ($p < 0.001$ for $CO_2$ and $ET$).

[Figure]

**Figure 10.** Gap-filled cumulative $CO_2$ (a) and evapotranspiration ($ET$) (b) fluxes of the agroforestry (AF) and grassland site during the period they were measured simultaneously in 2020. The red lines are the CON-EC fluxes and the light blue lines are the LC-EC fluxes. The dashed lines are the grassland site and the solid lines are the agroforestry site. The horizontal black dashed line in (a) indicates the transition of the ecosystem being either a $CO_2$ source (+) or sink (-).

**Minor comments**

1. The introduction is 6 paragraphs and could probably be a bit more concise. Eddy covariance isn't mentioned till the end of paragraph 2 and defined in paragraph 3. I would tend to combine everything before L32 into paragraph 1 (and write more compactly); perhaps L42-59 can also be more integrated and concise (e.g. the repetition of L51-52 – spectral...spectral, corrections...compensate; or L59 "which is an additional source of uncertainty in itself").

*We agree with your suggestion for the first two paragraphs, and reviewer #2 provided a similar comment for L19-33. Therefore, we rewrote, shortened and combined the first two paragraphs (L19-33) into one paragraph, which reads now as follows:*

Reducing carbon dioxide ($CO_2$) and other greenhouse gas (GHG) emissions can minimize the effects of global warming
20   and climate change (Griscom et al., 2017; Anderson et al., 2019; IPCC, 2021). In addition, mitigating $CO_2$ emissions with
Nature-based Climate (management) Solutions (NbCS) is seen as a fairly rapid and low-cost solution, which meanwhile can
provide environmental co-benefits (Griscom et al., 2017; Anderson et al., 2019). Agroforestry is an example of NbCS which can
contribute to resilient agriculture adapted for climate change, by providing a more favorable local microclimate (Schoeneberger
et al., 2012; Smith et al., 2013; Cardinael et al., 2021), increased biodiversity (Jose, 2009; Torralba et al., 2016), and a reduction
25   of soil erosion (Schoeneberger et al., 2012; van Ramshorst et al., 2022). Nevertheless, robustly validating estimations and
models of the carbon sequestration potential by NbCS is not straightforward and is time and labor intensive (Griscom et al.,
2017; Novick et al., 2022). Direct observations with eddy covariance (EC) can provide solid and independent measurements to
validate the carbon uptake of the entire ecosystem (Hemes et al., 2021; Novick et al., 2022; Wiesner et al., 2022).

*We understand the latter remark about repetition and tried to shorten the paragraph from L50-59 as follows:*

Spectral corrections are inevitable with the EC methodology (Massman and Clement, 2005; Emad, 2023), however the
additional loss of high-frequency signal of LC-EC setups increases the importance of these corrections applied (Mauder and
Foken, 2006; Reitz et al., 2022). Generally, the magnitude of spectral losses are for example depended on the response time
of sensors and the EC system as a whole (Leuning and Moncrieff, 1990; Massman and Lee, 2002; Polonik et al., 2019), the
measurement height of the EC tower (Moncrieff et al., 1997; Reitz et al., 2022), the length and diameter of the tubing when
present (Leuning and Moncrieff, 1990; Massman, 1991), the flow rate and flow regime inside the tube (Leuning and Moncrieff,
1990; Massman, 1991), and the absorption and desorption of water molecules inside the tubing (Massman, 1991; Ibrom et al.,
2007; Polonik et al., 2019). Furthermore, there are many different spectral correction methods available, each with their own
assumptions and uncertainties (Polonik et al., 2019; Reitz et al., 2022; Emad, 2023).

2.  L60 the introduction could finish with clearer hypotheses about the expected findings, rather than objectives only.

*We added a more explicit statement of the expected findings in the final paragraph of the introduction, which now reads as follows:*

In the current study we tested LC-EC setups over a temperate grassland and an adjacent alley cropping agroforestry grassland near Hanover in Germany. Due to the LC-EC setups larger loss of high-frequency signal, it is expected that the spectral corrections will be higher and more varied compared to the CON-EC. In order to identify the difference between the two EC setups, the objectives of this paper are to (i) perform a technical characterisation of the LC-EC setup relative to CON-EC in a temperate ecosystem setting, (ii) investigate the effect of the spectral correction method applied, and (iii) present the first application of LC-EC over a grassland and alley cropping agroforesty grassland.

3.  L90 or so could add specifications on precision of the instruments, not just their response times

*We added the precisions of the LC-EC instruments:*

measured with a HIH-4000 RH sensor (Honeywell International Inc., Charlotte, North Carolina, USA). The sensor response times of the GMP343 and HIH-4000 are 1.34 s and 4 s, respectively (Hill et al., 2017). The accuracy of the GMP343 and HIH-4000 are $\pm 5\ \mu mol_{CO2} \cdot mol^{-1}_{dry\ air} + 2\ \%$ of reading and $\pm 3.5\ \%$, respectively. The cell temperature ($T^{LC}_{CELL}$) was measured using a fine wire thermocouple (Omega Engineering Inc., Norwalk, Connecticut, USA) with a 0.2 s response time and $\pm 1.8$ K accuracy. The absolute cell pressure ($P^{LC}_{CELL}$) was measured using a MPX5100AP pressure sensor (NXP USA Inc., Austin, Texas, USA), with a $\pm 1.5$ kPa accuracy and 1 ms time response. The enclosure consists of a heater, which can reduce the

4.  L99 and 114 consider commenting on why the intake is 20cm below the sonic anemometer and risks of spectral attenuation since they would be sampling eddies at different wind velocities.

*Similar to reviewer #2: We placed the intake 20 cm below the center (at the bottom) of the sonic anemometer, so there is less disturbance of the wind measurements of the sonic anemometer. The effect of this sensor displacement is expected to be small, especially because it's below the sonic anemometer. The estimated flux loss is 0.71 % for the grassland site and 0.2 % for the agroforestry site, according to Eq. 27 from Kristensen et al. (1997). To clarify this in the paper we rephrased and added the following sentence to the LC-EC and CON-EC method section:*

was 0 m. By placing the intake at the bottom of the ultrasonic anemometer, the wind measurements are less disturbed, however small flux losses of 0.71% for the grassland tower and 0.2% for the agroforestry tower are expected based on calculations due to sensor displacement (Kristensen et al., 1997). The Synflex 1300 tube (1300-M0603, Eaton corporation, Dublin, Ireland)

5. L151 could add some brad description of the difference in theoretical background or implementation of the Horst and Ibrom and HL09 approaches

*Reviewer #2 had a similar comment, so accordingly we added some background information, especially focusing on the difference between the Horst and Ibrom method. This reads as follows:*

(2009) for sensor separation, were applied to investigate the sensitivity of the spectral correction method applied. The method of Horst (1997) is an analytical method, which uses a simple equation to estimate the spectral attenuation of each individual $CO_2$ and $H_2O$ measurement. The method of Ibrom et al. (2007) is an empirical method which is especially designed for the attenuation of the strongly RH depended $H_2O$ measurements in closed-path EC systems. This method defines the spectral attenuation on a large number of spectra based on RH humidity classes.

6. L198 have the residuals of the regression always been tested for normality?

*We plotted for the data from Fig. 4 the residuals vs. the fluxes (see figure below) and used a Shapiro-Wilk test. It shows that for $CO_2$ and LE the residuals are not normally distributed (Shapiro tests, p < 0.001). The LE residuals are increasing with increasing LE flux (clearly visible in the scatterplots from Figure 4, especially at the grasslands). The $CO_2$ flux residuals are also not normally distributed, which confirms the different regressions for positive and negative fluxes (section 3.2.2.). To explicitly state this in the paper, we added a Shapiro-Wilk test description to the method section 2.3.6. and in the result section 3.2.2. we added that the residuals are not normally distributed as follows:*

**2.3.6 Statistical methods**

Linear regressions were calculated by applying a major axis regression with R package *lmodel2* (Legendre and Oksanen, 2018) and the normality of the residuals was checked using Shapiro-Wilk normality tests with the R package *stats*. The root mean

**Section 3.2.2 scatter plots:**

generally lower than the CON-EC $CO_2$ fluxes, indicated by the slopes from linear regression below 1.0. The agreement for $CO_2$ fluxes between both EC setups was different for positive and negative fluxes, positive fluxes were overestimated (slope = 1.07–1.18) and negative fluxes were underestimated (slope = 0.86–0.96) (Table 3). This difference is also confirmed by the not normally distributed residuals of the linear regressions ($p < 0.001$). The correlation between the LE fluxes of both EC setups was lower compared to the $CO_2$ fluxes, especially for the grassland sites, which was also visible by the relatively large spread that increases with higher LE fluxes. This increasing spread is also confirmed by the not normally distributed residuals of the linear regressions ($p < 0.001$). Nevertheless, the slopes for the grassland and agroforestry campaigns in 2020 were good, 1.01

7. L206 and elsewhere – there are a lot of multiple "respectively" chains and perhaps this and other sentences like it can be simplified

*Thank you for the suggestion, but we prefer to keep this writing style.*

8. L219, 220, 224 all compare the systems or the sites, and the words "lower relative" or "higher" can be quantified with percent or absolute differences.

*We rewrote these sentences by adding absolute differences accordingly.*

*"The negative $CO_2$ fluxes during midday (8-17 h) of the LC-EC were on average 0.56 μmol $m^{-2}$ $s^{-1}$ lower relative to the CON-EC during all campaigns. The positive $CO_2$ fluxes of the LC-EC were similar to the CON-EC in all three campaigns."*

*"For example, the LE flux of the CON-EC at the agroforestry site was on average 18.4 W $m^{-2}$ higher compared to the LE flux of the LC-EC during the first 7 hours of the day, however this coincides with time periods when limited amount of data was available."*

9. L295 it's not clear to me – the most affected...by aliasing?

*Most affected by attenuation in the higher frequencies. We clarified this sentence as follows:*

higher frequencies. Moreover, the LC-EC $Co(wC_{CO_2})$ and $Co(wC_{H_2O})$ cospectra were quite similar for each setup, however the higher AF tower is the least attenuated and the grassland tower in 2021 is the most attenuated in the higher frequencies.

10. Fig 4 could include bias – both average and in different ranges (e.g. in the positive and negative FCO2 ranges separately)

*According to us, an indication of the bias between the two EC setups is already visible by the slope (deviation from 1.0). For positive and negative fluxes, the slope/bias is mentioned in the text. Nevertheless, based on the suggestion from reviewer #2 we included a table summarizing all the information more clearly in addition to Figure 4.*

**Table 3.** Additional statistics accompanying the scatter plots from Figure 4.

| $CO_2$ | r | n | RMSE ($\mu$mol m$^{-2}$ s$^{-1}$) | positive fluxes: slope ($R^2$); intersect | negative fluxes: slope ($R^2$); intersect |
|---|---|---|---|---|---|
| Grassland 2020 | 0.97 | 2726 | 1.9 | 1.07 (0.74); -0.48 | 0.86 (0.68); -0.64 |
| Agroforestry 2020 | 0.95 | 909 | 2.35 | 1.18 (0.53); -1.12 | 0.96 (0.69); 0.21 |
| Grassland 2021 | 0.98 | 2135 | 1.76 | 1.12 (0.68); -0.83 | 0.87 (0.81); -0.26 |
| **LE** | r | n | RMSE (W m$^{-2}$) | | |
| Grassland 2020 | 0.93 | 2270 | 130.99 | | |
| Agroforestry 2020 | 0.95 | 653 | 117.43 | | |
| Grassland 2021 | 0.92 | 1451 | 119.22 | | |

11. Fig 6 and likely elsewhere (e.g. Fig 7 and Table 3), "EC" is used rather than "CON-EC". I'd prefer the latter to be explicit and consistent throughout the text.

*You are completely right; this must have been forgotten when implementing "CON-EC" instead of "EC" previously. We updated Fig 6 & 7 and updated two tables.*

12. Fig 5a legend for LC-EC should be a green line; caption should have "light blue" as two words rather than one.

*We made changes accordingly (for the updated Figure see review #2)*

13. L345 consider more strongly highlighting that there is a 10-15% difference in results based on the correction method

*We quantified and included the difference between correction methods accordingly:*

the day (Figure 3). For the $ET$ measurements the difference between EC setups was on average 18.6% higher with the Ibrom method than with the Horst method (Table 7). In contrary, for the $CO_2$ fluxes the difference between EC setups was equal or

14. L462-470 could almost be in the intro to lead to hypothesis-driven objectives

*Similar to reviewer #2: We understand your point that this paragraph explains random white noise and aliasing (potentially methods), however we would like to leave it inside the spectral discussion to emphasize that the LC-EC (co)spectra look different compared to the CON-EC, but that this is not abnormal and an effect of the lower response time of the LC-EC sensors.*

15. L494 explain why the Bowen ratio decreased in these conditions

*We added an explanation why the Bowen ratio decreased, as follows:*

was not observed in the current study for either spectral correction methods (Figure 9). On the other hand, the studies of Barr et al. (1994) and Brotzge and Crawford (2003) measured and De Roo et al. (2018) modeled that the EBC decreased when the Bowen ratio decreased. The Bowen ratio decreases when the $\frac{ET_{actual}}{ET_{potential}}$ ratio increases (Eltahir, 1998). As 2021 was wetter and colder compared to 2020, the actual ET was closer to the potential ET and therefore the Bowen ratio was lower in 2021, which could explain that the LI-7200 performs worse in 2021. Additionally, Stoy et al. (2013) reports that wetlands, with likely

16. L537 be clearer on where and when this difference occurred

*We specified more clearly that the difference occurred over a one-month period.*

*"During simultaneous measurements at the agroforestry and grassland site, there was a significant difference in cumulative carbon uptake over a one-month period."*

17. L549 I think "lower in magnitude" and not in absolute number (since the LC system is higher at noon than the CON system in Fig 3 for CO2 flux)

*We changed this accordingly.*

**Technical comments**

Generally, there are many small issues in word order, punctuation and orthography, and vague writing that can be clarified and improved. I've noted some here.

1. L3 could add who this lower cost system is made by? (equivalent to L5 reference to Licor)

*Added (University of Exeter) to L3 and additionally added more details to the method section 2.2.1 on LC-EC.*

during the EC measurement campaigns in 2020 and 2021 was used for comparison. The LC-EC setup in the current study was very similar to the ones used by Cunliffe et al. (2022) and was custom built at the Department of Geography at the University of Exeter, United Kingdom. The LC-EC uses an uSONIC-3 Omni 3D ultrasonic anemometer (METEK GmbH, Elmshorn,

2. L8 no comma needed before seems

*Removed*

3. L11-12 is both vague and wordy (it's rather obvious that stronger attenuation requires larger spectral correction, and that thus the factors are amplified; perhaps there are better ways of saying this – more quantitative, more hypothesis oriented)

*We shortened this sentence and quantified the increased difference between spectral methods.*

2021 seems to be a consequence of the lower LE fluxes measured by the CON-EC. Due to the slower response sensors of the LC-EC setup, the (co)spectra of the LC-EC were more attenuated in the high-frequency range compared to the CON-EC. The stronger attenuation of the LC-EC led to larger cumulative differences between spectral methods 0.15–38.8%, compared to the CON-EC, 0.02–11.36%. At the agroforestry site where the flux tower was taller compared to the grassland, the attenuation was

4. L15 ET is never exactly 'equal' – perhaps "not statistically different" or similar

*This sentence is removed due to the additional gap-filling for Figure 11 and the rewriting of this paragraph.*

5. L19 no comma needed after climate change, "the" not needed before increased

*Removed both*

6. L32 consider "Direct observations with" before "eddy covariance"

*Included the suggestion*

7. L37 remove of

*Removed*

8. L77 the quotes should go after agroforestry and not project

*Changed*

9. L78 "if and" can be removed

*Removed*

10. L81 the comma after tower is not needed

*Removed*

11. L94 add s to time

*Added*

12. L104 "This" is ambiguous; in general it should be avoided without a noun following it

*Rephrased and made more explicit as follows:*

stainless steel 2 $\mu$m filters (SS-4FW-2, Swagelog, Solon, Ohio, USA). A nominal flow rate of $\sim 2$ L min$^{-1}$ was achieved with a NMP830KNDC-B diaphragm gas pump (KNF Neuberger Inc., Trenton, New Jersey, USA). The flow rate could drop down to $\sim 1$ L min$^{-1}$ when highly clogged. The flow rate resulted in a laminar flow with a Reynolds number of 717-358 inside the tubing (Massman, 1991).

13. L140 move "a" to after "narrow"

*According to us this a seems to be at the right place.*

expected that the time lag of $H_2O$ was at least equal or longer than the time lag of $CO_2$. In order to avoid a too narrow window for the time lag optimization in EddyPro, $\tau_{H_2O}^{max}$ was fixed at 40 s for all three campaigns and $\tau_{H_2O}^{min}$ was assumed equal to $\tau_{CO_2}^{min}$.

14. Table 2 could add tau-nom

*We added $\tau^{nom}$ to the table. Table 2 is now as follows:*

**Table 2.** Estimated time lag windows for $CO_2$ during each measurement campaign.

|  | Grassland 2020 | Agroforestry 2020 | Grassland 2021 |
|---|---|---|---|
| $\tau_{CO_2}^{min}$ (s) | 4.83 | 6.29 | 5.20 |
| $\tau_{CO_2}^{nom}$ (s) | 6.44 | 8.38 | 6.94 |
| $\tau_{CO_2}^{max}$ (s) | 9.66 | 12.57 | 10.41 |

15. L198 change were to was

*Instead we changed linear regression to linear regressions*

16. L230 "in the current paper" isn't needed.

*Removed and changed to "(figure not shown)" only*

17. L286, change look differently to looks different.

*Changed*

18. L320, 321, 325, consider "varied" instead of was/were varying

*Changed*

19. L360 remove also

*Removed*

20. L369 clarify "This" and consider changing "led to consistently."

*Sentence is rephrased and 'this' is made more explicit as follows:*

the slower response time of the LC-EC $CO_2$ and $H_2O$ sensors (Figure 6 & 7). The stronger attenuation of the LC-EC led to consistently higher spectral correction factors for the LC-EC setup compared with CON-EC (Figure 9).

21. L374 "which" would grammatically refer to the current LC-EC setup, but makes more sense in the sentence to refer to the predecessor. Reword sentence.

*Rephrased as follows:*

The study of Hill et al. (2017) compared a predecessor of the current LC-EC setup with an open-path LI-7500 IRGA at a 4.25 m tall tower on a pasture in Dumfries and Galloway, UK. This predecessor had a higher flow rate of approximately 75 L $min^{-1}$, but despite the different CON-EC IRGA and a higher flow rate, their results agree quite well with the current study.

22. L379 I think "led" not "lead"

*Changed to led*

23. L397 remove exact

*Removed*

24. Fig 11 caption – add "AF" after agroforestry (consider also defining ET)

*Added both to the figure caption*

25. L408 remove very

*Removed*

26. L414 care not carefulness

*Changed*

27. L420 remove comma after site

*Removed*

28. L489 could remind us how those results are poor

*Added explicitly what was poor as a reminder. The sentence is now as follows:*

The general characterisation of the LC-EC and CON-EC fluxes were discussed in the previous section, however the $H_2O$ flux of the CON-EC in 2021 will be discussed in more depth since the agreement between the LC-EC and CON-EC was poor.

29. L490 remove "of all"

*Removed*

30. L491 consider predicts instead of shows and removing probably

*We changed shows to predicts and removed probably*

31. L492 use either rather than both

*Changed*

32. L509-510 – be more concise

Shortened the sentence as follows:

It is not possible to point at a clear cause of the LE underestimations in 2021 and why this is not happening in 2020. It is clear that the difference in LE and EBC between the CON-EC and LC-EC increases with higher RH in 2021 (data not shown), which

33. L536 add such before as

*Added*

**Review #2**

Van Ramshorst et al. made a comparison of two eddy covariance setups using low-cost and conventional instrument(s) for a grassland and an agroforestry site in Germany. The authors showed that the low-cost EC system can measure fluxes at both landscapes. In general, the manuscript is of good scientific quality, it is well written, and fits to the scope of AMT. However, some issues need to be addressed before publication. I have major, minor, and technical comments which are listed below.

*Thank you for appreciating our study and constructive comments. Below you can find a point by point reply to your review.*

**Major comments:**

R1: In the abstract and conclusion it is written that low-cost EC system costs approximately 25% of the conventional EC system. Is this estimation made on material costs only or also working hours, building up the system, testing, setting up the logging software, etc.? How much time did you invest for preparation? The LICOR systems are standardized, have a robust measurement performance, and setting them up in the field may be less difficult than a custom-built setup. Then a customer may still buy the conventional instrument, which could also be the open-path $CO_2/H_2O$ gas analyzer of LICOR.

The authors can add a recommendation section with regard to preparation, costs, testing of software, and instruments of the LC EC setup. Estimates of the acquisition costs for the instruments could be added to Table 1. Summarizing the main aspects regarding flux pre- and post-processing may be also useful to add.

*We added an extra section to the discussion, elaborating on these suggestions and more clearly specified the costs reduction, this section reads as follows:*

**4.4 Costs of and considerations for a lower-cost eddy covariance setup**

The application of our LC-EC setup is less standardized compared to the more commonly used LI-7200. Nevertheless, the current study and the parallel study by Callejas-Rodelas et al. (2024) showed that LC-EC setups can be an alternative to CON-EC and provided elaborated examples on how to post-process the LC-EC data for other users. The post-processing of the LC-EC data requires some extra steps which are easy to implement, and the LC-EC flux calculations take approximately only 10% of the time compared to the CON-EC, due to the lower measurement frequency. The main advantage is the approximately 75% reduction in material costs (Cunliffe et al., 2022), as our LC-EC is approximately 11,000 euros, compared to more than 40,000 euros for the CON-EC with a LI-7200 (Callejas-Rodelas et al., 2024). Furthermore, the LC-EC setups have a lower power consumption, which makes them suitable for remote locations with only solar power available (Callejas-Rodelas et al., 2024). The LC-EC also requires maintenance and needs to be cleaned regularly, however calibrating the GMP343 with Vaisala software is straightforward and the HIH-4000 is long-lasting without calibration (Callejas-Rodelas et al., 2024). Finally, future LC-EC studies can contribute to further standardization and optimization of the employment and flux processing.

R2: The authors wrote that sonic measurements of CON EC were sampled at 2Hz instead of 20Hz during the 2020 campaigns (L279-292). This issue needs to be introduced earlier, preferably in the methods section. This introduces a significant uncertainty to the flux damping calculation. Instead of doing the flux analysis using 20 Hz data, it may be better to use the real 2Hz data of the CON EC setup for comparison.

*We understand this comment and similarly reviewer #1 suggested also to process the CON-EC in 2020 in 2 Hz instead of 20 Hz. Accordingly, we more explicitly mentioned the logging error in the method section and emphasized that the CON-EC data in 2020 is in 2 Hz. All the 2020 fluxes and spectra are recalculated and this (mostly) improved the agreement (slopes) between the LC-EC and CON-EC in 2020, as visible below in the scatterplots. Furthermore, we updated all figures, results and discussion using the 2 Hz CON-EC fluxes/results in 2020.*

[Figure]

R3: I think you shouldn't write that the LC system ''potential to measure EC fluxes at various land use systems'' (L16 and L560). Fluxes are reported only for two ecosystems. Also, we learn that the performance of $CO_2$ flux measurements is better than for latent heat fluxes, and a competitive performance was achieved above tall canopies. I think this should be the final remark. It can be added to the conclusion that existing research infrastructures equipped with EC setups like ICOS can be used for further validation of the instrument setup and testing of flux damping methods. This can also be an aspect for a recommendation section.

*We agree that in the current study we only measured at a grassland and agroforestry site, however in the parallel study of Callejas-Rodelas et al. (2024) LC-EC fluxes at a cropland performed also well compared to CON-EC. We changed a variety of land uses to grassland and agroforestry inside the conclusions and abstract. Additionally, we added the possibility to do measurements in other ecosystems for further development of the LC-EC. The final paragraph of the conclusions now reads as follows:*

Finally, the results show that LC-EC has the potential to measure EC fluxes at a grassland and agroforestry system for approximately 25% of the costs of a CON-EC system. The performance of the $CO_2$ flux is better than the LE flux and at the taller agroforestry tower the results are more consistent. The LC-EC setups can be used to increase the spatial representativeness of flux measurements in heterogeneous ecosystems. Design-wise a shorter and heated inlet tube would be recommended and additional LC-EC characterisation studies could take place at a variety of ecosystems with CON-EC setups (e.g., ICOS, FLUXNET). These future in-depth investigations could also lead to further optimization of the spectral corrections.

**Minor comments:**

R4: In general, the manuscript is a technical study. The ecological aspect is kept short in the discussion, which is fine. However, the reader gets an introduction about the potential benefits of agroforestry regarding climate change (L19-33). I think this section can be shorter.

*We agree with your suggestion, and reviewer #1 provided a similar comment. Therefore, we rewrote, shortened and combined the first two paragraphs (L19-33) into one paragraph, which reads now as follows:*

Reducing carbon dioxide ($CO_2$) and other greenhouse gas (GHG) emissions can minimize the effects of global warming and climate change (Griscom et al., 2017; Anderson et al., 2019; IPCC, 2021). In addition, mitigating $CO_2$ emissions with Nature-based Climate (management) Solutions (NbCS) is seen as a fairly rapid and low-cost solution, which meanwhile can provide environmental co-benefits (Griscom et al., 2017; Anderson et al., 2019). Agroforestry is an example of NbCS which can contribute to resilient agriculture adapted for climate change, by providing a more favorable local microclimate (Schoeneberger et al., 2012; Smith et al., 2013; Cardinael et al., 2021), increased biodiversity (Jose, 2009; Torralba et al., 2016), and a reduction of soil erosion (Schoeneberger et al., 2012; van Ramshorst et al., 2022). Nevertheless, robustly validating estimations and models of the carbon sequestration potential by NbCS is not straightforward and is time and labor intensive (Griscom et al., 2017; Novick et al., 2022). Direct observations with eddy covariance (EC) can provide solid and independent measurements to validate the carbon uptake of the entire ecosystem (Hemes et al., 2021; Novick et al., 2022; Wiesner et al., 2022).

R5: The intakes of both setups were 20 cm below the center of their sonic anemometer (L99 and L113). The influence of the vertical separation on flux damping should be discussed. (Kristensen et al., 1997).

*Similar to reviewer #1: We placed the intake 20 cm below the center (at the bottom) of the sonic anemometer, so there is less disturbance of the wind measurements of the sonic anemometer. The effect of this sensor displacement is expected to be small, especially because it's below the sonic anemometer. The estimated flux loss is 0.71 % for the grassland site and 0.2 % for the agroforestry site, according to Eq. 27 from Kristensen et al. (1997). To clarify this in the paper we rephrased and added the following sentence to the LC-EC and CON-EC method section:*

relative humidity inside the measuring cell during humid conditions, to prevent condensation. The vertical separation between the center of the ultrasonic anemometer and the intake of the sampling tube was -0.2 m and the East- and Northward separation was 0 m. By placing the intake at the bottom of the ultrasonic anemometer, the wind measurements are less disturbed, however small flux losses of 0.71 % for the grassland tower and 0.2 % for the agroforestry tower are expected due to sensor displacement (Kristensen et al., 1997). The Synflex 1300 tube (1300-M0603, Eaton corporation, Dublin, Ireland) had a length of either 2

R6: To determine minimum and maximum lag, the nominal lag was multiplied with 0.75 and 1.5, respectively (L137-L138). How were these numbers determined?

*This range was manually determined based on the distribution of the time lags. EddyPro uses a ± range which is similar for the lower and upper limit (e.g.: 0.75 and 1.25), but the distribution of the LC-EC time lags isn't equally distributed, as clearly shown now in Table 2 (see R7). Due to the longer tubing in combination with the lower flowrate the time lags are often larger, during for example (partly) clogging or (in combination with) high RH inside the tubing. We highlighted why the time lag is different a bit clearer in section 2.3.1 "pre-processing" point 5.*

5. The time lags of the LC-EC systems in the current study were considerably larger and more variable compared to a CON-EC setup with a LI-7200, due the longer tubing in combination with a lower flow rate. This led to unsatisfactory time lag optimization when the standard time lag estimation method in EddyPro was applied. Therefore, realistic time

R7: Were values different from 40 s tried as upper limit for $H_2O$ flux calculation (L141)? Did you apply a time lag filter? What is the time lag range of quality-assured fluxes?

*Yes, originally we used the same time lag window estimation method for the $H_2O$ flux as for the $CO_2$ flux, however this resulted in many time lag estimates at the upper limit (1.5\*nom). As a result, this led to worse agreement between the LE fluxes of the LC-EC and CON-EC. By extending the upper limit to 40s a better distributed time lag estimation was obtained and this resulted in a better agreement for the LE fluxes between the LC-EC and CON-EC. To give an indication of the time lag ranges of the quality-assured fluxes we added two extra rows to Table 2 in section 2.3.1 as follows:*

to $\tau_{CO_2}^{min}$. In Table 2, the estimated time lag ranges for quality controlled $CO_2$ and $H_2O$ fluxes calculated by EddyPro are shown for the adapted time lag estimation.

**Table 2.** Estimated time lag windows for $CO_2$ during each measurement campaign and time lag ranges for quality controlled $CO_2$ and $H_2O$ fluxes calculated by EddyPro.

|  | Grassland 2020 | Agroforestry 2020 | Grassland 2021 |
|---|---|---|---|
| $\tau_{CO_2}^{min}$ (s) | 4.83 | 6.29 | 5.20 |
| $\tau_{CO_2}^{nom}$ (s) | 6.44 | 8.38 | 6.94 |
| $\tau_{CO_2}^{max}$ (s) | 9.66 | 12.57 | 10.41 |
| Time lag range $CO_2$ (s) | 5.0 – 8.0 | 6.0 – 12.0 | 5.0 – 9.5 |
| Time lag range $H_2O$ (s) | 5.0 – 28.5 | 7.0 – 33.0 | 6.0 – 29.0 |

R8: Please add references for each of the preprocessing steps (L147).

*We added explicit references to specific steps in the pre-processing section, as follows:*

were pre-calculated, as described in pre-processing step 2, 3 & 4 in Section 2.3.1.

as shown in pre-processing step 5 in Section 2.3.1, were applied.

R9: A theoretical background should be added to the methods of Horst et al. (1997) and Ibrom et al. (2007) (L165-L166). Please check line 164. Low-frequency corrections are described in Moncrieff et al. (2004).

*We have added some background information for both method, so the main difference between the methods becomes clear. This reads as follows:*

(2009) for sensor separation, were applied to investigate the sensitivity of the spectral correction method applied. The method of Horst (1997) is an analytical method, which uses a simple equation to estimate the spectral attenuation of each individual $CO_2$ and $H_2O$ measurement. The method of Ibrom et al. (2007) is an empirical method which is especially designed for the attenuation of the strongly RH depended $H_2O$ measurements in closed-path EC systems. This method defines the spectral attenuation on a large number of spectra based on RH humidity classes.

*Furthermore, Moncrieff et al. (1997) is changed to Moncrieff et al. (2004)*

R10: Absolute limits for flux filtering were applied. Are the flux limits based on manual screening? If so, please add it.

*We explicitly added that the absolute limits for the fluxes were based on manual screening of the data:*

*"Furthermore, absolute limits for the $CO_2$, LE and H fluxes were applied, based on manual screening of the data."*

R11: Figure 2: Please adjust the x-axis. It's easier to compare the campaigns if both x-axes have the same starting date and change day of year to dates which are easier to read like Aug 15. The distance between the plot labels, (a) and (b), y-axis can be increased a little bit.

*We adjusted Figure 2 accordingly*

[Figure]

R12: Figure 3: Move the titles in the middle. The distance between the plot labels, (a) and (d), y-axis can be increased a little bit.

*We have updated the figure accordingly*

[Figure]

R13: Figure 4: Put the same titles on top of each row. There's a lot of information in the plot. I propose to summarize the information in a table.

*We updated the titles inside the figure, and only left the linear interpolation results inside the figure. Additionally, we created a table with more detailed information and statistics, also including suggestions by reviewer #1. The text in 3.2.2. is slightly rewritten to state less numbers.*

[Figure]

**Table 3.** Additional statistics accompanying the scatter plots from Figure 4.

| $CO_2$ | r | n | RMSE ($\mu$mol m$^{-2}$ s$^{-1}$) | positive fluxes: slope ($R^2$); intersect | negative fluxes: slope ($R^2$); intersect |
|---|---|---|---|---|---|
| Grassland 2020 | 0.97 | 2725 | 1.9 | 1.07 (0.74); -0.48 | 0.86 (0.68); -0.64 |
| Agroforestry 2020 | 0.95 | 909 | 2.35 | 1.18 (0.53); -1.12 | 0.96 (0.69); 0.21 |
| Grassland 2021 | 0.98 | 2135 | 1.76 | 1.12 (0.68); -0.83 | 0.87 (0.81); -0.26 |
| **LE** | r | n | RMSE (W m$^{-2}$) | | |
| Grassland 2020 | 0.93 | 2269 | 130.68 | | |
| Agroforestry 2020 | 0.95 | 653 | 117.43 | | |
| Grassland 2021 | 0.92 | 1451 | 119.22 | | |

R14: Figure 5: Same as for figure 4. Also, the sections 3.2.2 and 3.2.3 are difficult to read because a lot of numbers are mentioned in text. A table can be helpful here. Align the format of the x-axes of (b), (d), and (f). Start and end date should be the same.

*We updated the titles inside the figure, aligned the x-axes of the left and right column figures and only left the slope results inside the figure. The start and end date of the right column figures can't be the same as all three campaigns have different time periods and lengths.*

*As in section 3.2.2. (R13), we summarized the results of the EBC into a table in section 3.2.3. and slightly rewrote the text to state less numbers. The EBR ratio results are also summarized, but in another table in section 3.3.*

[Figure]

**Table 4.** Energy balance closure (EBC) for both EC setups and for all three campaigns.

| EBC | LC-EC | | CON-EC | |
|---|---|---|---|---|
| | r | slope ($R^2$) | r | slope ($R^2$) |
| Grassland 2020 | 0.90 | 0.85 (0.81) | 0.95 | 0.83 (0.90) |
| Agroforestry 2020 | 0.91 | 1.01 (0.83) | 0.92 | 0.99 (0.85) |
| Grassland 2021 | 0.91 | 0.83 (0.83) | 0.96 | 0.75 (0.93) |

R15: Sec. 3.2.5: Maybe put this section to the method part? The influence of the concentration correction on fluxes is not in the discussion.

*We moved this section to the methods part as suggested.*

R16: Figures 6 and 7: Add the ensemble averaged (co)spectra to (e) and (a) and add titles to each row. How many (co)spectra were used for averaging? Explain the inertial subrange in the method section. Choose a different color for the temperature (co)spectra. They are difficult to distinguish.

*For Fig. 6 and 7 we have added ensemble averaged (co)spectra and titles to each row and changed the color of the LC-EC temperature (co)spectra. We defined the inertial subrange in section 3.2.4 by "In general, the spectra of the LC-EC show a stronger decay in energy content compared to the spectra of the CON-EC in the higher frequency range (i.e. inertial subrange)...". We created an additional table stating the number of $CO_2$ or $H_2O$ (co)spectra used for the ensemble averages in Fig. 6 and 7.*

[Figure]

**Table 5.** Number (n) of $CO_2$ and $H_2O$ (co)spectra used for the ensemble averages of the LC-EC and CON-EC in Figure 6 & 7.

| ($n_{LC-EC}$ ; $n_{CON-EC}$) | Spectra | | | Cospectra | |
|---|---|---|---|---|---|
| | $CO_2$ | $H_2O$ (50 %) | $H_2O$ (80 %) | $CO_2$ | $H_2O$ |
| Grassland 2020 | 1332 ; 1271 | 232 ; 291 | 29 ; 114 | 1332 ; 1271 | 794 ; 1113 |
| Agroforestry 2020 | 697 ; 689 | 70 ; 89 | 31 ; 35 | 697 ; 689 | 235 ; 304 |
| Grassland 2021 | 703 ; 683 | 90 ; 111 | 30 ; 85 | 703 ; 683 | 320 ; 524 |

R17: Figure 8: is the figure needed? Since it is not referred in the discussion, put it in Appendix or supplement.

*We removed this figure and it will be added as supplement.*

R18: Sec 3.3: Put the three observations in a list, which makes the text easier to read.

*We created a list with these observations.*

R19: Figure 9: Put titles on top of each row and align the format of the x-axes and if possible also of the y-axes. Maybe the latter only for CO₂. Make a space between g and C in y-label.

*We have added the titles to each row and reformatted the y-axis for the CO₂ plots. Aligning the x-axis is not possible due to different measurement dates as mentioned in an earlier comment, this would create a lot of white space and would to worse readability. The space is added to the y-label.*

[Figure]

R20: Figure 10: Put titles on top of each row. The labels (a) to (f) look kind of large compared titles and labels. In the legend, the identifier for box plots, median and whisker, are hardly visible. Put numbers on top of each box showing how many values were used. Make the box borders, medians, and whiskers more visible. You may want to leave out the outliers. Then you see the range of the boxes better.

*We have added titles to each row and decreased the size of the labels. The legend keys are increased for better readability. The number of measurements used for the four boxplots of each site are added to the figure (for each site all 4 boxplots use the same dataset size). We left out the outliers and improved the boxplot readability by increasing the linewidths.*

[Figure]

R21: Figure 11: Make a space between g and C in y-label.

*Added the space in the y-label*

R22: Sec. 4.1: The beginning of section 4.1 is a summary and not a discussion. (L360-370)

*We agree that the beginning was too much of a summary, however we think it is good to start the discussion with highlighting the message that the LC-EC is able to reproduce the CON-EC fluxes. After that the comparison with other studies and more discussion starts. We shortened this paragraph and it now reads as follows:*

**4.1 Technical characterisation**

The current study showed that the LC-EC was able to capture the diel pattern and ecosystem response of the $CO_2$ and LE fluxes observed at the grassland and agroforestry grassland by the CON-EC. The stronger attenuation of the LC-EC led to consistently higher spectral corrections for the LC-EC setup compared with CON-EC (Figure 9). Nevertheless, the LC-EC setup showed a strong correlation with the CON-EC, with $r = 0.95$–$0.98$ and $r = 0.92$–$0.95$ for the $CO_2$ and LE fluxes, respectively (Figure 4). The LC-EC $CO_2$ flux was slightly lower compared to CON-EC, indicated by the linear regression slopes of $0.93$–$0.96$ ($R^2 = 0.91$–$0.95$). The LC-EC LE fluxes in 2020 were slightly higher compared to CON-EC, indicated by linear regression slopes of $1.01$ and $1.05$ ($R^2 = 0.85$–$0.9$), and have similar diel cycles. The LE fluxes in 2021 did not agree well, and this observation will be discussed elaborately in section 4.1.

R23: L397: The unpublished work by Callejas-Rodelas et al. (2023) …. I can't find Callejas-Rodelas et al. (2023) in the references. If a manuscript is not published, it's better to write in the references a remark *in submission* or *in discussion* instead of always repeat unpublished work. Anyway, I think it's published now: https://www.sciencedirect.com/science/article/pii/S0168192324002016

*You are correct, we updated all these references with the now published paper by Callejas-Rodelas et al. (2024): https://www.sciencedirect.com/science/article/pii/S0168192324002016*

R24: L462-470: Maybe move this to the introduction or method section?

*Similar to reviewer #1: We understand your point that this paragraph explains random white noise and aliasing (potentially methods), however we would like to leave it inside the spectral discussion to emphasize that the LC-EC (co)spectra look different compared to the CON-EC, but that this is not abnormal and an effect of the lower response time of the LC-EC sensors.*

**Technical comments:**

L4: comma after study

*Added*

L19: no comma

*Removed*

L20: leave out the "-" between carbon and dioxide.

*Removed*

L32 and L31: both sentences start with Eddy covariance.

*One of the sentences is adapted according to the suggestion of reviewer #1. The two sentences read now as follows:*

(Griscom et al., 2017; Novick et al., 2022). Direct observations with eddy covariance (EC) can provide solid and independent measurements to validate the carbon uptake of the entire ecosystem (Hemes et al., 2021; Novick et al., 2022; Wiesner et al., 2022).

Eddy Covariance is a non-invasive technique to directly measure the net land-atmosphere exchange (flux) of energy, water ($H_2O$), $CO_2$ and other GHGs over an area of up to several hectares (Baldocchi, 2003; Lee et al., 2005; Baldocchi, 2008). Cur-

L36-L37: add Heiskanen et al. (2022) as reference.

*Added as reference next to Sabbatini et al. (2018) and Pastorello et al. (2020)*

L37: no comma after impacts

*Removed*

L60: comma after study

*Removed*

L72 and L74: Both sentences start with based on gap-filled meteorological data. Consider changing.

*We slightly rephrased one of the two sentences and they now read as follows:*

9.6 °C; based on the Hanover weather station of the German Meteorological Service (station ID: 2014). Based on gap-filled meteorological data of our own grassland site in Mariensee, in 2020 and 2021 the annual precipitation was 521 mm and 597 mm, and the annual mean temperature was 11.3 °C and 9.8 °C, respectively. The long term mean wind speed at 3.0 m height was 1.87 m s$^{-1}$ and the dominant wind directions at the site were west and southwest, based on gap-filled meteorological data of Mariensee from 2019-2021.

L81: remove the tall after 3m. 10m sounds tall but not 3m.

*Changed to 3 m in height*

L94: add s to time

*Added*

L106: comma after 2021

*Added*

L170 and L172: make $CO_2$ straight.

*Changed*

L186: put brackets around 1.

*Added*

L191: make H italic.

*Changed*

L194: comma after however

*Added*

L198: add s to regression.

*Added*

L203 and L204: comma after 2020 and 2021

*Added both*

L213: put bar on top of $u*$. The bar refers to an average.

*Changed $u*$ to $\overline{u*}$*

L229: H is straight here. Change to italic. Also at other occurrences.

*Changed occurrences of H to H.*

L230: figure not shown is sufficient

*Changed.*

L234: Use a large ''-'' which makes clearer that a range is meant here. Same for all occurrences.

*Changed all ranges in the paper to a large "-"*

L246: $r$ or $R^2$?

*Small r is correct*

L257: change biggest to largest (same in L262) and add ''the'' after between

*Changed both*

L266: comma after in general.

*Added*

L314 and L315: comma after campaign.

*Added*

L353, 354, and 355: repetition of similar. Consider rephrasing.

*This section is rewritten because of the additional gap-filling (review #1)*

L356: change has to had and sites to site.

*Changed*

L357: is there a which between ratio and was missing?

*Seems correct as is.*

L407: replace big by large.

*Changed*

L421: remove ''friction velocity'' and commas

*Removed*

L422: comma after air

*Added*

L427: remove tall after 3 m.

*Removed*

L477: put brackets around 1.

*Added*

L486: confirm instead of confirms.

*Changed*

L499: had instead of has

*Changed*

L530: closed-path

*Added*

L549: comma after 2021

*Added*

**References:**

Heiskanen, J., Brümmer, C., Buchmann, N., Calfapietra, C., Chen, H., Gielen, B., Gkritzalis, T., Hammer, S., Hartman, S., Herbst, M., Janssens, I., Jordan, A., Juurola, E., Karstens, U., Kasurinen, V., Kruijt, B., Lankreijer, H., Levin, I., Linderson, M.-L., Loustau, D., Merbold, L., Lund Myhre, C., Papale, D., Pavelka, M., Pilegaard, K., Ramonet, M., Rebmann, C., Rinne, J., Rivier, L., Saltikoff, E., Sanders, R., Steinbacher, M., Steinhoff, T., Watson, A., Vermeulen, A., Vesala, T., Vítková, G., and Kutsch, W.: The Integrated Carbon Observation System in Europe, B. Am. Meteorol. Soc., 103, E855-E872, https://doi.org/10.1175/BAMS-D-19-0364.1, 2022.

*Added*

Kristensen, L., Mann, J., Oncley, S. P., and Wyngaard, J. C.: How Close is Close Enough When Measuring Scalar Fluxes with Displaced Sensors?, J. Atmos. Ocean. Tech., 14, 814–821, https://doi.org/10.1175/15200426(1997)014<0814:HCICEW>2.0.CO;2, 1997.

*Added*

Moncrieff, J. B., Massheder, J. M., deBruin, H., Elbers, J., Friborg, T., Heusinkveld, B., Kabat, P., Scott, S., Soegaard, H., and Verhoef, A.: A system to measure surface fluxes of momentum, sensible heat, water vapour and carbon dioxide, J. Hydrol., 188, 589–611, https://doi.org/10.1016/S0022-1694(96)03194-0, 1997.

*Was kept included*

Moncrieff, J. B., Clement, R., Finnigan, J., and Meyers, T.: Averaging, Detrending, and Filtering of Eddy Covariance Time Series, Kluwer Academic, Dordrecht, 7–31, https://doi.org/10.1007/14020-2265-4_2, 2004.

*Added*